# S-Glutathionylation and S-Nitrosylation in Mitochondria: Focus on Homeostasis and Neurodegenerative Diseases

**DOI:** 10.3390/ijms232415849

**Published:** 2022-12-13

**Authors:** Sofia Vrettou, Brunhilde Wirth

**Affiliations:** 1Institute of Human Genetics, University Hospital of Cologne, University of Cologne, 50931 Cologne, Germany; 2Center for Molecular Medicine Cologne, University of Cologne, 50931 Cologne, Germany; 3Institute for Genetics, University of Cologne, 50674 Cologne, Germany; 4Center for Rare Diseases, University Hospital of Cologne, University of Cologne, 50931 Cologne, Germany

**Keywords:** glutathionylation, nitrosylation, redox, mitochondria, neurodegeneration, Alzheimer’s, Parkinson’s, amyotrophiclateralsclerosis, Friedreich’s ataxia

## Abstract

Redox post-translational modifications are derived from fluctuations in the redox potential and modulate protein function, localization, activity and structure. Amongst the oxidative reversible modifications, the S-glutathionylation of proteins was the first to be characterized as a post-translational modification, which primarily protects proteins from irreversible oxidation. However, a growing body of evidence suggests that S-glutathionylation plays a key role in core cell processes, particularly in mitochondria, which are the main source of reactive oxygen species. S-nitrosylation, another post-translational modification, was identified >150 years ago, but it was re-introduced as a prototype cell-signaling mechanism only recently, one that tightly regulates core processes within the cell’s sub-compartments, especially in mitochondria. S-glutathionylation and S-nitrosylation are modulated by fluctuations in reactive oxygen and nitrogen species and, in turn, orchestrate mitochondrial bioenergetics machinery, morphology, nutrients metabolism and apoptosis. In many neurodegenerative disorders, mitochondria dysfunction and oxidative/nitrosative stresses trigger or exacerbate their pathologies. Despite the substantial amount of research for most of these disorders, there are no successful treatments, while antioxidant supplementation failed in the majority of clinical trials. Herein, we discuss how S-glutathionylation and S-nitrosylation interfere in mitochondrial homeostasis and how the deregulation of these modifications is associated with Alzheimer’s, Parkinson’s, amyotrophic lateral sclerosis and Friedreich’s ataxia.

## 1. Introduction

Reactive oxygen and nitrogen species (RONS) are byproducts of aerobic metabolism and constitute essential signaling molecules [1]. The RONS signaling cascade is mediated by the oxidation–reduction (redox)-based post-translational modifications (PTMs) of proteins [2].

The mammalian brain is a major source of RONS due to its high metabolic activity [1]. Consequently, redox PTM-mediated signaling is pivotal for the brain under normal physiology [3]. In contrast, under pathological conditions, aberrantly produced RONS cause oxidative and/or nitrosative stress, which, in turn, damages DNA, proteins and lipids and leads to impaired cellular function [4,5]. The mitochondrial oxidative phosphorylation (OXPHOS) system and enzymes such as NADPH oxidases (NOXs) and nitric oxide synthases (NOSs) are the main sources of RONS in aerobic organisms [6,7,8]. Some major RONS are hydrogen peroxide (H_2_O_2_), hydroxyl radical (OH), superoxide anion (O_2_), nitric oxide (NO) and peroxynitrite (ONOO^−^) [9]. Concomitantly, there is a tightly modulated machinery that reduces the levels of RONS in the cell, which is constituted by small molecules (ascorbate, cysteine, glutathione (GSH) and proteins glutathione peroxidases (GPxs), catalases (CATs), superoxide dismutases (SODs) and peroxiredoxins (PRDXs)) [10,11,12,13]. Under eustress (normal physiological conditions), there is a balance between RONS production and reduction, whereas any imbalance in this system causes oxidative/nitrosative stresses [4,11]. The complex crosstalk between reactive oxygen species (ROS) and reactive nitrogen species (RNS) triggers a downstream cascade of events that collectively are termed redox signaling [4,5,11].

The main targets of redox PTMs are proteins due to their high abundance and increased rate constants for oxidation reactions [11,14,15,16]. While redox PTMs can happen on almost every amino acid, redox modifications of thiol happen only on cysteine (Cys) residues. Cys residues constitute the primary drivers of redox signaling due to their high rates of oxidation, which are mostly reversible, and reduction [17]. Cys residues can be targets of multiple redox PTMs due to the electronic structure of the thiol (-SH) group, which allows for multiple oxidation states (ranging from −2 to +6) [17]; for example, protein disulfide isomerase (PDI) can be both S-nitrosylated and S-glutathionylated at its reactive thiols [18]. The redox PTMs of Cys in proteins have multiple effects on the particular protein (Figure 1). Under eustress, only a subset of redox-sensitive Cys residues within proteins is susceptible to multiple redox PTMs, whereas, under oxidative and/or nitrosative stress, even lowly reactive Cys residues undergo multiple redox PTMs, which can be excessive and/or irreversible [19]. Redox PTMs in proteins might act as a “redox-switch” that modifies the protein’s function: for instance, rendering an enzyme active or inactive [20]. Redox PTMs can alter the protein structure via disulfide bond formations, Cys-dependent metal cofactors interactions and the modulation of the topography of the protein [17,21]. Moreover, redox PTMs can modify the stability and degradation of the target protein, which depends on the oxidation status, which is reversible or irreversible, via ATP- or ubiquitin-dependent or independent mechanisms [22,23]. Finally, the redox status of a protein can affect its localization and translocalization: for instance, by changing the conformation of a protein and rendering it available for interactions with other proteins that translocate into other subcompartments [17,24]. Collectively, this continuum of the reactive Cys residues of proteins constitutes a redox-sensing mechanism with which the cell reacts to the differential level of RONS and adjusts to its redox status via multiple downstream effects (Figure 1).

Aberrant redox PTMs, including S-glutathionylation and S-nitrosylation, can alter multiple processes within the cell and have been extensively studied, mainly separately, in mitochondria homeostasis and neurodegenerative diseases [25,26,27,28,29]. Of particular importance and extensively reviewed in the following sections is the trigonal interaction of oxidative/nitrosative stress, mitochondrial dysfunction and neurodegeneration. We incorporated both S-glutathionylated and S-nitrosylated proteins in this review, with a particular focus on mitochondrial proteins or proteins that directly affect mitochondria and that have been extensively investigated (or their role is still elusive) in neurodegenerative disorders, including Alzheimer’s disease (AD), Parkinson’s disease (PD), amyotrophic lateral sclerosis (ALS) and Friedreich’s ataxia [30,31,32,33].

The aim of this review is to bring together and compare S-glutathionylation and S-nitrosylation processes that regulate mitochondria and interfere with or exacerbate neurodegenerative disorders. By doing so, the converging redox-induced mechanisms between neurodegenerative disorders might unravel commonalities in neurodegeneration or aging processes and, thus, contribute to novel therapeutic approaches beyond antioxidant supplementation, which has failed in clinical trials [34]. While the focus was mainly on AD, PD, ALS and Huntington’s disease in the majority of other reviews that had the same purposes, we focused on neurodegenerative disorders where mitochondria dysfunction and S-glutathionylation/S-nitrosylation have been proposed to interfere with the disease’s pathogenesis. In addition, we bring into focus Friedrich’s ataxia, which is a mitochondrial neurodegenerative disorder in which nitric oxide-mediated S-nitrosylation signaling has not yet been investigated, and we urge the research field to shed light on that aspect of the disorder by discussing the data that indirectly indicate its possible importance. The major objectives of this review are to re-introduce redox PTMs as ROS/RNS-induced cell signaling processes with pivotal roles in mitochondria homeostasis and neurodegeneration and as converging mechanisms between homeostasis, aging and neurodegenerative disorders. Given the significant leap in our scientific knowledge about their multiverse roles in the last two decades, particularly in neurodegenerative disorders, we aimed herein to combine previous and new data demonstrating not only their independent roles but also their interdependent function that orchestrates mitochondria regulation and triggers or exacerbates neuronal degeneration.

This review has some limitations concerning the extent of biochemical principles governing the multiple ways that can lead to S-glutathionylation and S-nitrosylation of proteins. For this reason, we cite appropriate literature for the particular cases where biochemical processes are not extensively described. Despite that, the topics covered in this review focus on signaling pathways for which the information stated is adequate for the reader to comprehend thoroughly the cellular processes implicated in mitochondria homeostasis and disease.

## 2. Protein S-Glutathionylation

Protein S-glutathionylation (PSSG) is a reversible oxidative post-translational modification that is mediated by the conjugation of glutathione (GSH) to an exposed Cys residue (-SH). The S-glutathionylation of mitochondria proteins is highly sensitive to local fluctuations in redox status, particularly via the alterations of the ratio between reduced and oxidized glutathione (GSH/GSSG) [35]. Proteins can be S-glutathionylated either non-enzymatically or enzymatically (Figure 2). Non-enzymatic protein S-glutathionylation takes place via different mechanisms that have been extensively reviewed by many authors [35,36,37]. Non-enzymatic protein S-glutathionylation is very non-specific and primarily occurs under oxidative stress. In mitochondria under oxidative stress, proteins can be hyper-glutathionylated, and this effect can be utilized as an oxidative stress marker [38]. Under normoxia, a number of mitochondrial proteins are S-glutathionylated at specific Cys residues in a reversible, sensitive-to-redox-fluctuations and enzymatically driven manner. S-glutathionylation reactions have been particularly investigated in cytosol where glutaredoxin 1 (GRX1) resides while being the chief enzyme responsible for PSSG de-glutathionylation [39]. The mitochondrial GRX1 isozyme, GRX2, is only ~34% identical to the sequence of GRX1 but shares the exact same catalytic mechanism [40]. The sequence differences between GRX1 and GRX2 are important for their regulation. GRX2, unlike GRX1, has no exposed thiol, making it less susceptible to ROS-mediated deactivation. Moreover, GRX2 forms a homodimer that is stabilized by a 2Fe-2S cluster, for which its disassembly by ROS superoxide, O_2_^−^, leads to the release of active monomeric GRX2. This regulation potentiates the link between S-glutathionylation and mitochondrial ROS [41]. Both GRX2 and GRX1 can mediate S-glutathionylation, but the exact mechanisms are still elusive. GRX2 has been documented in mediating Complex I (CI) and UCP3 S-glutathionylation and de-glutathionylation; the significance of these reactions will be discussed in the following sections [42,43]. It is also worthy of mention that Glutathione-S-transferases (GST Pi and Mu) have been shown to perform S-glutathionylation in cytosol and GST Alpha, Kappa, Mu, Pi and Zeta have been proposed to have similar functions in mitochondria [44,45]. The small oxidoreductase, sulfiredoxin (SRX), catalyzes the reduction of sulfinic acid derivatives into two Cys peroxiredoxins; therefore, it has been proposed to have deglutathionylation activity, too [46] (Figure 2).

While there are many sources for mitochondrial ROS, mitochondrial regulations by redox switches depend on mitochondrial densities, protein thiols contents, the import of GSH and export of GSSG and the reduction of GSSG to GSH by glutathione reductase [42,47]. In the following sections, we will discuss S-glutathionylation in mitochondria’s core processes and its effect on various proteins and enzymes, highlighting the impact on mitochondrial metabolism and function.

### 2.1. Regulation of OXPHOS System by S-Glutathionylation

Most information about how redox switches affect mitochondria thiols has been accrued by the multiple studies on the impact of S-oxidation in the OXPHOS system (Figure 3). The CI of the mitochondrial electron transport chain (ETC) was the first OXPHOS component that was identified to be S-glutathionylated. The oxidation of cysteine residues on the 51-kDa and 75-kDa subunit in CI can be protected from irreversible oxidation by S-glutathionylation [48,49]. While the S-glutathionylation of CI decreases its activity, this modification is reversible by deglutathionylation via GRX2 [42]. Another important role of the S-glutathionylation of CI is that it diminishes O_2_^−^ production by sterically blocking the NADH’s binding site [50]. However, the other S-glutathionylation of CI has also been shown to increase O_2_^−^ emissions but only under prolonged S-glutathionylation [51]. It is important to correlate the S-glutathionylation of CI with the S-glutathionylation of the α-ketoglutarate dehydrogenase complex (KGDH) since KGDH produces the NADH that is utilized for oxidation by CI and both enzymes produce ROS. Both the S-glutathionylation of KGDH and CI diminish their activity under excess H_2_O_2_. Once the levels of H_2_O_2_ and O_2_^−^ are back to normal, CI and KGDH are deglutathionylated by GRX2 and are active again [52]. In contrast to CI, Complex II (Succinate dehydrogenase (SDH) S-glutathionylation increases the activity of the enzyme. In particular, CII is persistently S-glutathionylated at the 70-kDa FAD binding subunit; thus, this constitutive modification is considered to be optimal for the CII’s activity [49]. Complex V (ATP synthase, α subunit) has also been shown to be glutathionylated, and this modification diminishes ATP production by blocking nucleotide binding [53]. The temporary glutathionylation of CV might act synergistically with CI and KGDH glutathionylation upon increased levels of ROS. Additionally, this modification diminishes ATP hydrolysis and, consequently, proton leaks, thus preventing the depolarization of the mitochondrial inner membrane [54].

### 2.2. Regulation of Nutrient Metabolism by S-Glutathionylation

The tricarboxylic acid cycle (TCA) is the most central mechanism for both anaerobic and aerobic organisms. In aerobic organisms, the TCA cycle provides the essential building blocks for macromolecules and strips electrons from nutrients in order to provide ATP. The impact of TCA cycle enzyme S-glutathionylation is enzyme-dependent (Figure 3). Aconitase (ACN) can be reversibly S-glutathionylated at its 4FE–4S cluster upon H_2_O_2_ bursts [55]. However, this modification does not disassemble its 4FE–4S cluster due to its interaction with the Frataxin protein [55]. Only upon excessive stress is ACN irreversibly deactivated due to the 4FE–4S cluster’s disassembly [28]. NADP-dependent isocitrate dehydrogenase (IDH), which contributes to the NADPH pool in mitochondria, can be inactivated by S-glutathionylation and that might hamper its reducing power, which is necessary in order to preserve the antioxidant’s defense system [56]. As mentioned above, KGDH can be glutathionylated and reversible oxidation might have a dual role, which includes the modulation of the flow of metabolites via the TCA cycle and amino acid metabolism and the accumulation of 2-oxoglutarate, which in turn quenches H_2_O_2_ and minimizes H_2_O_2_ production [57]. Pyruvate dehydrogenase (PDH) can be also regulated by reversible S-glutathionylation on its E2 subunit in a similar manner to KGDH [58]. It has been reported that excessive S-glutathionylation inhibits lipid oxidation, thus decreasing fatty-acid-supported OXPHOS [59]. More specifically, mitochondrial long-chain fatty acid oxidation demands the mitochondrial import of long-chain fatty acyl carnitine esters. CACT is an antiporter that is impeded in the mitochondrial inner membrane, which exchanges acyl-carnitines for L-carnitine from the matrix [60]. When the ratio of GSH/GSSG is low, the capacity of CACT for exchange is inhibited by S-glutathionylation [61]. Collectively, both KGDH and PDH are supposed to be redox sensors, and their capacities to commit carbon to further oxidation are redox-regulated. In a similar manner, malate dehydrogenase (MDH) and succinate-CoA translocase (SDT) are both inactivated by S-glutathionylation [62]. These modifications in the OXPHOS system and TCA cycles link the modulation of nutrient delivery and oxidation in mitochondria to S-glutathionylation reactions (Figure 3).

### 2.3. Regulation of Mitochondrial Permeability Transition Pore and Apoptosis by S-Glutathionylation

It has been reported that uncoupling proteins (UCPs) modulate ROS release via proton leaks in mitochondria, supporting the hypothesis that they assist in diminishing oxidative stress and modulating the relationship between ROS and fuel metabolism [63,64]. Several studies demonstrated that UCP2 and UCP3 (mostly expressed in muscle) are modified by S-glutathionylation, which affects ROS levels [62,64]. The S-glutathionylation of UCP2 and UCP3 leads to increased ROS production, whereas deglutathionylation increases mitochondrial respiration, diminishes ROS production and lowers mitochondrial membrane potential [65]. In the particular case of UCP3, the importance of its glutathionylation was exemplified by the discovery that GRX2 is required for its modification upon its increase in membrane potential and high levels of mitochondrial ROS [43]. Mitochondria tightly regulate the cell’s fate by harboring several death factors, including the apoptogenic factor cytochrome c [66]. Amongst the many ways that mitochondria can facilitate cell death, one involves the opening of mitochondrial permeability transition pores (MPTPs). MPTP opening is induced by many stressors and high ROS, serving as a redox sensor, for which its modulation can be attributed to the S-glutathionylation of various components of the pore [67]. For instance, the S-glutathionylation of ANT has been shown to prevent MPTP opening, whereas the decreased S-glutathionylation of ANT is associated with MPTP opening and apoptosis in neurological tissues [68]. Similarly, the S-glutathionylation of Cyclophilin D (CYPD) prevents mitoptosis by blocking the pore’s opening. Mitochondria suicide (mitoptosis), described in detail by Skulachev in the early 21st century, defines the multiple ways in which mitochondria are degraded without extramitochondrial components [69]. The initial triggers of mitoptosis are excessive ROS, potent inducers of MPTP opening, which prime mitochondria for mitoptosis and degradation by inner or outer mitochondria swelling [69]. In addition, the S-glutathionylation of CYPD might prevent MPTP openings during mitochondria-to-cell communication, and it connects redox buffering with cell signaling in this manner [70]. Interestingly, not many studies have characterized the glutathionylation events inside the mitochondria’s inter-membrane space (IMS) in humans. On the one hand, the import of proteins to the mitochondrial matrix relies on a disulfide-exchange relay system; on the other hand, GRX1 has been identified to reside in IMS, pinpointing the observation that S-glutathionylation may play a crucial role in IMS [70]. However, a recent study showed that the human S-glutathionylation of MIA40, an IMS import receptor, facilitates the maintenance of ROS levels and the optimal function of CIII and CIV [71]. This mechanism highlights the interconnection between ROS levels, protein S-glutathionylation and mitochondrial respiration (Figure 3).

### 2.4. Regulation of Mitochondria Morphology by S-Glutathionylation

Fusion and fission processes tightly regulate mitochondria morphology and cell fate and are mediated by GTPases located at the outer mitochondrial membrane (OMM). Fusion is regulated by the GTPases mitofusins (MFNs) 1 and 2 and the autosomal dominant optic atrophy protein 1 (OPA1), which also modulate mitochondria cristae [72]. Fission is primarily initiated by the dynamin family of GTPases (DRP1) [72]. Under chronic oxidative and nutrient stresses, fission is triggered, leading to mitochondrial fragmentation. In contrast, under acute stress and glutathione depletion, the GSSG-induced S-glutathionylation of MFN2 and OPA1 oligomerization results in mitochondrial hyperfusion [73]. It has been attributed a role in adaptation to an acute stressor in the mitochondrial hyperfusion process, which provides protection from excessive oxidation and at the same time consolidates ATP production capacities [74]. Therefore, the hyperfusion state induced by S-glutathionylation of MFN2 can act as a regulatory mechanism under acute stressors, serving as a transient response that provides a critical window for adaptation while preserving ATP production (Figure 4). Recently, it was shown that S-glutathionylation of MFN2 led to the formation of necrosomes and their recruitment to mitochondria-associated endoplasmic reticulum (ER) membranes (MAMs), and in turn, it induced ER-mitochondria intra-organelle crosstalk-dependent neuronal necroptosis [75]. This was the first identification that MFN2 S-glutathionylation increases sensitivity to the neuronal necroptotic mechanism in vitro and in vivo. This mechanism can therefore describe an effect of mitofusin’s S-glutathionylation that goes beyond mitochondrial deregulation and can affect other organelles with detrimental consequences to cell fate.

## 3. Protein S-Nitrosylation

Non-canonical nitric oxide (NO) signaling encompasses covalent post-translational modifications of biomolecules via NO and NO derivatives. These modifications include the S-nitrosylation of protein thiols, metal nitrosylation of transition metals and oxidative nitration or hydroxylation of various molecules such as tyrosine, amines, fatty acids, etc. [76]. Concerning tyrosine nitration, the formation of peroxynitrite (ONOO^−^) and nitrogen dioxide (NO_2_), are associated with this irreversible modification that can impact protein function. In the case of cysteine residues, peroxynitrite triggers the formation of oxygenated forms such as sulfenic acid (-SOH), sulfinic acid (-SO_2_H) and sulfonic acid (-SO_3_H) [77]. S-thiolation forms including S-glutathionylation are induced by peroxinitrite and nitrosothiol formation [77]. Nitrosylation is the reaction in which a nitrosyl moiety of NO is incorporated into another molecule, and when this affects the thiol group of Cys, the reaction is termed S-nitrosylation [78]. S-nitrosylation can propagate beyond the boundaries of its localization via transnitrosylation [33], and in recent years, it has emerged as a prototype redox-induced cell-signaling mechanism that protects cells against oxidative stress and diminishes ROS levels [79]. NO is not an antioxidant and cannot strongly react with protein thiols. Therefore, in order for the majority of S-nitrosylation reactions to take place, NO needs to react with oxygen to increase its oxidation state and then react with protein thiols [80]. For this reaction, many mechanisms have been proposed [76].

Many factors can modulate S-nitrosylation levels including the redox state. More specifically, the decreased levels of antioxidants, which increase the oxidation potential of the cells, trigger the S-nitrosylation of proteins, whereas increased levels of antioxidants have the opposite effect [81]. In particular, under glutathione depletion, the S-nitrosylation of mitochondrial proteins is elevated in order to protect cells against thiol oxidation, membrane permeabilization and apoptosis [82]. Another factor that modulates S-nitrosylation is the denitrosylation process. While the S-nitrosylation process is, in general, a non-enzymatic process (with an exception for prokaryotes), the opposite reaction, denitrosylation, can be enzymatic or non-enzymatic [83]. The spontaneous cleavage of the S-nitrosyl group can be performed by metal ions, nucleophiles, reducing agents, heat, ROS and UV [83]. Enzymatically driven denitrosylation can be processed by two major denitrosylase systems in cells: S-nitrosoglutathione reductase (GSNOR) and thioredoxin reductase (TRXR). Of particular importance for cell homeostasis is the balance between S-nitrosylation and denitrosylation. For example, the over-expression of GSNOR in the brain has been associated with cognitive impairment [84]. Transnitrosylation on the other hand, which is the transfer of the nitrosyl moiety of a nitrosylated protein to the reactive Cys of the interacting protein, allows for signal transmissions, even in sites that are distant from the initial NO source [33]) (Figure 5).

In mitochondria bioactivities and quality control, S-nitrosylation and denitrosylation have an integral role [85,86]. The S-nitrosylation of mitochondrial thiols is influenced by fluctuations in mitochondrial respiration and redox states [87]. Intriguingly, the source of NO inside the mitochondria is still controversial. One hypothesis is that mitochondria harbor a particular mitochondria NOS isoform (mtNOS) [88], while another suggests that NO enters mitochondria from cytosol via transnitrosylase-mediated transportation [35]. In the following sections, we will discuss multiple mitochondrial targets of S-nitrosylation, which when modified by NO, mostly inhibits their activity to modulate ROS production, O_2_ consumption and mitophagy, while simultaneously providing protection against cell death [88].

### 3.1. Regulation of the OXPHOS System by S-Nitrosylation

All mitochondrial complexes CI-V can be S-nitrosylated and, in turn, inhibited in order to minimize ROS production under oxidizing conditions [89]; moreover, it scavenges NO, prohibiting it from reacting with superoxides to produce peroxynitrite [90] (Figure 3). In hypoxic conditions, NO production is elevated, which could again lead to the S-nitrosylation and inhibition of CI-V enzymatic activities [89]. Under this condition of diminished O_2_ consumption and the redistribution of intracellular oxygen, the stabilization of hypoxia-inducible factor-1α (HIF1A) can be prevented during hypoxia [91]. The inactivation of CI-V ultimately impedes mitochondrial respiration, which in turn causes mitochondrial depolarization [92]. Consequently, the PINK1/PARKIN pathway (described in detail later) is triggered and induces mitophagy to protect cells from oxidative stress by eliminating defective mitochondria [93]. Additionally, relative to CI-V, cytochrome c, which transfers electrons from CIII to CIV, is also a target of S-nitrosylation. Specifically, cytochrome c is nitrosylated at the iron metal center of the heme moiety, which then transfers the nitrosyl group to glutathione, resulting in the formation of the endogenously abundant transnitrosylase GSNO [94]. While S-nitrosylated cytochrome c is released outside mitochondria during apoptosis, it has been proposed that it can suppress apoptosis by retaining its capability to synthesize GSNO [95].

### 3.2. Regulation of Metabolic Enzymes by S-Nitrosylation

The fatty acid β-oxidation and aerobic oxidation of pyruvate (TCA cycle) to CO_2_ take place at the inner mitochondrial membrane and matrix. These processes not only generate ROS by themselves [96,97] but also produce the source of electrons and cofactors that are utilized by the ETC, which in turn also produces ROS [96,98]. Because of this, under oxidative stress, S-nitrosylation inhibits multiple enzymes in these two catabolic processes to inhibit their activities in order to minimize ROS production [87] (Figure 3). Concerning the ETC and TCA cycle, these enzymes include aconitase, the α-ketoglutarate dehydrogenase complex and succinate dehydrogenase, while in the case of fatty acid catabolism, long-chain and short-chain acyl-CoA dehydrogenase and enoyl-CoA hydratase, carnitine palmitoyl transferase 2 and flavoprotein dehydrogenase belong to the enzymes that are inhibited [87]. A recent study revealed a novel mechanism with which RNS generated by macrophages can render inactive α-ketoacid dehydrogenases (including pyruvate dehydrogenase complex and α-ketoglutarate dehydrogenase complex) [99]. In this study, it was shown that the lipoid arm of α-ketoacid dehydrogenase can be modified by RNS-mediated thiol modifications, and in turn, the acyl-transfer activity of the lipoid arm is blocked leading to the inhibition of pyruvate dehydrogenase complex and α-ketoglutarate dehydrogenase complex.

### 3.3. Regulation of Apoptosis by S-Nitrosylation

Under oxidative conditions, the S-nitrosylation of ETC complexes and the TCA cycle inhibits their activities and the consequent production of ROS, while under the same conditions, S-nitrosylation inhibits pro- and anti-apoptotic or antinecrotic proteins to protect cells from death. This bidirectional S-nitrosylation-mediated inhibition assures that upon decreased energy production, death pathways are impeded (Figure 3). The mitochondrial pro- and anti-apoptotic proteins include the following:Mitochondrial permeability transition pores (MPTPs) are formed under oxidative stress and modulate the redox potential. Under oxidative stress, MPTP increases the permeability relative to high molecular weight macromolecules, leading to mitochondria swelling and necroptosis [100]. Under the basal levels of NO, cyclophilin D (CYPD), critical for an MPTP opening, can be S-nitrosylated [101]. This inhibits its interaction with MPTPs, preventing pore opening and protecting cells during stress. Conversely, under the increased production of NO, peroxynitrite is being produced by excessive ROS, which leads to the opening of MPTPs, the ablation of ATP production and necrosis [102].VDAC has multiple roles. Its localization at the mitochondrial outer membrane allows VDAC to regulate the fluxes of metabolites. At the same time, VDAC modulates cytochrome c, which activates caspase-dependent apoptotic cell death [103]. Similarly to MPTP regulation by NO, VDAC S-nitrosylation at basal levels of NO inhibits its function, protecting cells from apoptosis, whereas elevated levels of NO upregulate VDAC functions [104].Crucial to their role in apoptosis, caspase-3 and caspase-9 are S-nitrosylated in the absence of apoptotic triggers, while their denitrosylation leads to their activation upon the activation of the Fas receptor [105].Another component of the outer mitochondrial membrane with an anti-apoptotic role involves BCL2 family proteins, which interfere with apoptosis by regulating the release of cytochrome c [106]. Upon apoptotic signaling, the S-nitrosylation of BCL-2 inhibits its ubiquitination and degradation, allowing it to exert anti-apoptotic protection [107].

### 3.4. Regulation of Fission by S-Nitrosylation

A recent discovery in neurons showed that the mitochondrial fission protein DRP1 is regulated by NO. It was shown that the NO-mediated S-nitrosylation of DRP1 triggers its GTPase activity and fission [108]. This discovery is of particular importance for neurons, where the nNOS-mediated regulation of DRP1 might contribute to mitochondrial fragmentation, followed by synaptic loss and neuronal cell death, which are observed in neurodegenerative disorders such as AD [109]. This mechanism of the S-nitrosylation-mediated regulation of fission contrasts the S-glutathionylated regulation of fusion, underlying the importance of these two redox PTMs in modulating mitochondria homeostasis (Figure 4).

## 4. Redox PTMs in Alzheimer’s Disease 

### 4.1. S-Glutathionylation in Alzheimer’s Disease

Alzheimer’s disease (AD) is a progressive neurodegenerative disorder and the most prevailing form of dementia [110]. AD is characterized by the progressive loss of memory and cognitive function and brain changes that include brain atrophy, amyloid-beta (Aβ) peptide build-up and extracellularly hyperphosphorylated tau proteins, which form neurofibrillary tangles (NFTs) and, intracellularly, resulting in neuronal death and inflammation [111]. The majority of AD patients present the sporadic form of AD (sAD), which corresponds to 95% of AD cases, whereas only 5% of all cases account for the familial form of AD (fAD). Age is the main risk factor for developing AD, as well as genetic factors including apolipoprotein E (*APOE*) and triggering receptor expressed on myeloid cells 2 (*TREM2*); moreover, acquired risk factors such as metabolic co-existing conditions are also main risk factors for developing AD [110]. The predominant sporadic forms of Alzheimer’s and Parkinson’s disease are the most common forms with unknown etiologies [112]. While both AD and PD are progressive neurodegenerative disorders that affect the elderly and are characterized by distinct histopathological hallmarks based on patients’ postmortem tissues, they share common molecular and cellular characteristics. In particular, the trigonal interaction between proteinopathy, oxidative stress and mitochondrial dysfunction represent cardinal features in both disorders [112]. The prevailing theory that explains this interaction suggests that the expression and protein aggregation of Aβ in AD (and alpha-synuclein in PD) is modulated by reactive oxygen species and transition metal ions [112]. Consequently, the formed oligomers or aggregates lead to various kinds of mitochondrial damage, such as impaired bioenergetics, fusion and fission and mitophagy, as was described in isolated mitochondria, cell cultures and postmortem brain tissues. In turn, these dysfunctional mitochondria lead to excess ROS production that triggers cellular death pathways [113]. Given the myriad interactions between oxygen radicals, mitochondria and toxic protein aggregates, the elucidation of the exact mechanistic pathway that triggers this trigonal feedback loop could lead to a better understanding of the commonalities between aging processes and neurodegenerative disorders as well as better biomarkers, prognosis and treatment [112].

#### 4.1.1. S-Glutathionylation/Glutathione as Potential Biomarkers for Alzheimer’s Disease

During aging, oxidative stress is increased in the brain due to the imbalance of the redox status, which includes the production of excess ROS or the deregulation of the antioxidant defense system [114]. Accruing evidence strongly suggests that oxidative stress can temporally precede the onset of AD pathogenesis [115]. An essential factor of AD-related oxidative stress could be attributed to the glutathione regulation of oxidative stress [115]. Many groups independently described the depletion of glutathione and, particularly, a decreased ratio of GSH/GSSG in AD patients’ erythrocytes and in patients with mild cognitive impairments in the blood and brain, which precedes AD [116,117]. Additionally, glutathione depletion was negatively correlated with brain amyloidosis in the temporal and parietal regions, which was observed in a study assessing brain glutathione in AD patients using MRI [118]. These findings indicate the multifaceted role of glutathione as a biomarker in the early stages of AD and further strengthen the rational of glutathione depletion related to oxidative stress in the formation of amyloid plaques. Given the amount of data underlying a faulty glutathione metabolism, many studies characterized the protein levels and activity of glutathione-related enzymes. Several attempts to measure the enzymatic activities of GPX and GR in AD patients provided controversial results [115]. In some studies, GR has been reported as decreased [119], while in others, there was no change. Concerning GPX activities in AD, in some cases, it was found to be decreased [120], whereas, in others, it increased [119,121]. To add more complexity to these discrepancies, conflicting results have been provided from the measurement of GSH levels in AD patients’ postmortem tissues, which can be attributed to the quality of the tissues. However, this accumulation of evidence renders the usage of antioxidant enzymes, or solely the ratio of GSH/GSSG, inadequate when utilized as prognostic markers of AD [115]. In contrast, there are emerging data not only showing that the level of glutathionylated proteins provides a valuable indication for AD’s staging but also provides valuable information for the underlying molecular phenomena that contribute to disease pathogenesis. In severely affected regions of the AD brain, redox proteomic approaches provided evidence that there are increases in S-glutathionylated proteins in the inferior parietal lobe of AD patients compared to the control groups [122]. In other studies, it was proposed that S-glutathionylation might represent an early step before AD progression [123]. There were reports suggesting that Aβ accumulation and oxidative stress do not exist solely in neuronal cells but also exist in peripheral blood. The use of a combination of electrophoresis and principal component analysis of S-glutathionylated proteins in the blood and brain from AD transgenic mice of different disease stages and non-AD mice allowed the prediction of early-stage AD [124]. Another group recently provided the first evidence reporting that *APOC1* and *TOMM40* polymorphisms might represent independent risk factors for developing AD, and its major variants are correlated with the disruption of biothiol metabolism and the insufficient removal of DNA oxidation [125]. The glutathione in mitochondria (mtGSH) represents a separate cytosolic glutathione pool, which inhibits intramitochondrial proteins in exiting the cytosol and can induce a cell-death cascade [126]. Since glutathione synthesis takes place in the cytosol, specific carriers such as TOMM40 are required for glutathione to enter mitochondria. The underexpression and blockade of TOMM40 resulted in impaired glutathione translocations into the mitochondria, which leads to increased ROS production, mitochondria failure and caspase-mediated cell death, contributing to neuronal degeneration and the onset of dementia [127]. Importantly, similarly to glutathione, which utilizes the TOMM40 machinery to enter mitochondria, the DNA repair enzyme 8-oxoguanine DNA glycosylase (OGG1) needs TOMM40 to translocate inside the mitochondria [128]. Early in the course of AD, the accumulation of 8-oxo-2′-deoxyguanosine (8-oxo2dG) in DNA has been detected and its formation has been attributed to increased mitochondrial ROS production [129]. The authors correlated specific polymorphisms in *APOE*4, *TOMM40*′650 and *APOC1*′623 with biothiol measurements (homocysteine and glutathione levels) in the plasma of AD patients and controls, providing an explanation for the development of AD in young age patients with inordinate oxidative stress responses due to *TOMM40* and *APOC1* rare variants [125]. 

#### 4.1.2. Redox and Metabolic Enzymes in Alzheimer’s Disease

Aside from the importance of the S-glutathionylation profile as a biomarker of disease progression or prognosis in AD, the elucidation of the identity of glutathionylated proteins in AD provided significant information for the molecular phenomena that underlie the disease’s progression (Figure 6). The first identity of the differential S-glutathionylated proteins was uncovered by Dalle-Donne et al., in which case glutathionylated actin had impaired function [130]. Later, Newman et al. identified the exact targets of S-glutathionylation in the inferior parietal lobe of AD patients compared to the control. The authors demonstrated that two glycolytic enzymes, GAPDH and α-enolase, had increased levels of S-glutathionylation, which decreased their activity. Deoxyhemoglobin was the third target of S-glutathionylation, which decreases its ability to supply oxygen to the neurons in the brain and likely decreases energy production. The fourth target, α-Crystallin B, is a heat-inducible chaperone that is co-deposited with Aβ plaques and its oxidation leads to dysfunction and potential contribution to protein misfolding in AD progression [122]. During aging, the impaired control of ROS impacts mitochondria, in particular, the major source of ROS in cells, and results in the oxidation of sulfhydryl groups in mitochondria proteins. In AD brains, the diminution of α-ketoglutarate dehydrogenase (KGDH) activity of the TCA cycle has been observed, which can impair glucose utilization [131]. KGDH has been characterized as a major mitochondrial redox sensor modulating its activity via S-glutathionylation to control ROS production; at the same time, it is one of the few targets of mitochondrial GRX2 [132]. Whether the S-glutathionylation of KGDH decreases its activity in AD and, thus, impairs glucose utilization and ROS production is worthy of investigation. The expression and redox status of GRX1 and GRX2 has been well characterized in AD models and patients. GRX1 and GRX2 expression showed a significant decrease in the axonal area of the hippocampus of CA1 of AD compared to control patients, with no differences in neuronal cell bodies [133]. Interestingly, in SH-SY5Y cells treated with Aβ, the increased oxidation of GRX1 was observed with a consequent activation of pre-apoptotic signaling kinase ASK1 that decreased the viability of the cells, whereas the overexpression of GRX1 rescued the viability of Aβ-treated SH-SY5Y cells [134]. Additionally, overexpressing GRX1 in the brains of APP/PS1 mice not only restored memory recall after contextual fear conditioning but also reversed F-actin loss in spines [135]. Thus, increasing GRX1 levels might be a potential therapeutic treatment for AD.

#### 4.1.3. Role of Glutathione in Aβ and Tau Accumulation in Alzheimer’s Disease

Another implication of glutathione depletion is associated with the increased oxidation of M35 residues in Aβ peptides leading to lipid peroxidation, the formation of amyloid plaques and, consequently, neurofibrillary tangles [136]. In mitochondria, the intracellular accumulation of Aβ is correlated with mitochondria dysfunction and more specifically with defects in the TCA cycle, the diminished activity of CIII and CIV of ETC and increased ROS [137]. While the exact mechanism of Aβ-induced mitochondria toxicity has not been mechanistically elucidated, Chen and Jan identified that Aβ binds to Aβ-binding alcohol dehydrogenase (ABAD) inside mitochondria and this interaction leads to mitochondrial impairment [137]. Concerning protein misfolding inside mitochondria, studies in the AD frontal and temporal regions of the brain confirmed an increased expression of cytosolic and mitochondrial human branched-chain aminotransferase proteins (hBCATs) and suggested an increased response to cellular redox disturbances in assisting neural protection [138]. hBCATs catalyze the reversible transamination of the alpha-amino group of the branched-chain amino acids into α-ketoglutarate, forming their respective branched chain α-keto acids and glutamate [139]. While the physiological role of the redox-active CXXC motif of hBCAT is unclear, Hindy et al. demonstrated a novel functional role in protein folding. In particular, they showed that the hBCAT mitochondrial isoform colocalizes with protein disulfide isomerase (PDI) and MIA40 inside the mitochondria in the human brain and that the S-glutathionylation of mitochondrial hBCAT plays a protective role in preserving hBCAT function during cellular stress [138] (Table 1). This finding potentiates the role of S-glutathionylation not only in protecting redox-sensitive proteins but also points out that it is integral in protein folding catalyzed by hBCAT. Both hBCAT and PDI are involved in AD; further understanding their redox regulation in protein folding will pave the way for targeted therapeutical interventions in AD. Another role for S-glutathionylation in AD involves the polymerization of S-glutathionylated tau (or G-S-S-Tau) into AD-like paired-helical filaments (PHFs) [140] (Table 1). The authors showed that S-glutathionylated tau (C-terminal microtubule-binding region) might prevent filament assembly in vitro, potentiating the crucial role of this property during the early stages of oxidative stress [140]. Later, it was shown that, under long-term mitochondrial stress, cells treated with NAC (precursor of glutathione) and MitoQ exhibited the diminution of tau dimerization, underlying the key role of ROS-mediated tau aggregation under mitochondrial stress [141] (Figure 6). Altogether, these data provide evidence for a dual role of S-glutathionylation in the major hallmark manifestations of AD. On the one hand, S-glutathionylation of hBCAT, which colocalizes with Aβ in mitochondria, might indirectly regulate protein misfolding [138]. On the other hand, S-glutathionylation of tau has a direct effect on the probability of tau forming filaments, thus connecting oxidative stress with tau misfolding [140].

#### 4.1.4. GSTs Diverse Roles in Alzheimer’s Disease

During the last few years, a novel multifaceted role of a key enzyme of glutathione metabolism, glutathione-S-transferase, has been introduced in AD. Glutathione-S-transferases (GSTs) belong to a family of phase II enzymes, which catalyze the conjugation of a wide variety of electrophilic substrates including xenobiotics (pesticides, chemicals, drugs and carcinogens) to glutathione, and thus represent a potential risk factor for AD [167]. The enzymatic inactivity of GSTs is correlated to higher oxidative stress [168]. In postmortem brain tissues and the CSF of AD patients, decreased GST activities have been reported [169]. In a study conducted on Brazilian people, the genetic polymorphisms of GSTP1 (V allele) and the nullity of GSTT1 proved to be additional risk factors for late-onset AD together with APOE polymorphism ε4, whereas the polymorphism of GSTM1 had no discriminations between groups [170]. In a subsequent study, Iranian people with APOE ε4 and GSTM1 null deletions had an increased risk for late-onset AD [171]. In AD, anxiety and depression have been hypothesized to be early-onset symptoms of AD, with the latter representing the most common psychiatric disorder in AD with a potential treatment to remain elusive [172]. Anxiety-like behavior is closely related to oxidative stress [173]. Melatonin, an endogenous hormone that decreases during aging and in AD, has been shown to improve memory loss and learning in 3xTg-AD mice as well as to decrease Aβ accumulation and tau hyperphosphorylation in various AD models [174,175]. Based on this knowledge, Nie et al. treated 3xTg-AD and WT mice with melatonin and performed proteomics in their corresponding hippocampus regions [176]. Their data showed that between the differentially expressed proteins, glutathione S-transferase P 1 (GSTP1) (anxiety-associated protein) and complexin-1 (CPLX1) (depression-associated protein) was significantly downregulated in the hippocampus of 3xTg-AD mice compared with the WT mice. The expression of these two proteins was modulated by melatonin treatments, suggesting that melatonin could be a potential treatment of neuropsychiatric symptoms in AD via the modification of GSTP1 and CPLX1 [176]. Another novel role of GSTs and the glutathione system and relevance to neurodegenerative disorders has been identified in a recent study in fly wing axons and primary rodent neurons, where it was uncovered that the glutathione redox pathway regulates mitochondria dynamics in axons [177]. In particular, this study in *Droshophila* unraveled a novel GST, Gfzf, homologous to GSTT1 in humans, that inhibits mitochondrial hyperfusion under normal conditions and that changes the redox status due to GST loss, having a direct impact on mitochondrial trafficking and neuronal responses. The authors demonstrated that GSTs constitute novel components of the mitochondrial fusion inhibition machinery in vivo. Therefore, they potentiated the key redox regulatory differences between compartments in neurons (axons vs. soma) and highlighted the importance of the further investigation of the ratio of GSH:GSSG and GST activities in the mitochondria’s dynamic-driven axonal loss in neurodegenerative diseases [177].

### 4.2. S-Nitrosylation in Alzheimer’s Disease

Investigations with respect to S-nitrosylated proteins in AD brain tissues provided evidence for pervasive S-nitrosylation dysregulation in AD, suggesting that this redox post-translational modification is critical in disease pathogeneses [178]. Both ROS and RNS are implicated in metabolic dysfunction and protein misfolding in AD. In particular, NO-induced post-translational modifications have been extensively studied in mitochondrial bioenergetics and dynamics as well as in Aβ and tau aggregates in AD [179].

In a study that utilized targeted proteomics for the detection of S-nitrosylated proteins in different regions of the AD brain (hippocampus, substantia nigra and cortex), 45 endogenous nitrosylated proteins were identified with main functions in signaling pathways, metabolism, apoptosis and redox regulation [30] (Figure 7). Of particular interest for the mitochondrial homeostasis and metabolism in AD were the differentially nitrosylated proteins: Mn superoxide dismutase (SOD2), fructose-bisphosphate aldolase C (ALDOC) and voltage-dependent anion-selective channel protein 2 (VDAC2) (Table 1). Interestingly, the authors assessed the functional interaction of the identified nitrosylated proteins and demonstrated that the mitochondrial VDAC2, which was hypernitrosylated, particularly interacted with VDAC1 and cytoskeletal proteins such as tubulin and ACT. VDAC1’s abnormal interaction with amyloid beta and phosphorylated tau has been reported to block mitochondrial pores leading to mitochondrial dysfunction [180]. The movement of Ca^2+^ across the outer mitochondrial membrane has been suggested to be mediated by VDACs [144]. Since the amyloid beta abnormal modification of transported Ca^2+^ and buffering systems has been shown to elevate levels of intracellular Ca^2+^ [181], it can be speculated that the S-nitrosylation of VDACs affects their roles in Ca^2+^ uptake and signaling and thus contributes to Ca^2+^ dysregulation in mitochondria. Concerning the interaction between VDAC and cytoskeletal proteins, presuming that S-nitrosylation is a priming factor in protein folding [85], the observed nitrosylation of tubulins (TUBA-1A, 1B chain and TUBB-2C) and their interaction with ACT and hypernitrosylated VDAC2 might be a crucial factor that hampers microtubule architecture in AD and other neurodegenerative diseases. Another mitochondrial protein identified to be S-nitrosylated in the AD region is SOD2; its nitrosylation might inhibit its detoxifying capacity, leading to mitochondrial dysfunction [143] (Table 1). It is still uncertain how the S-nitrosylation of SOD2 contributes to AD pathology.

Importantly, the key genetic factor for late-onset AD, APOE, can be nitrosylated (isoforms e2 and e3) both in cell cultures and human hippocampi tissues, suggesting a potential role of S-nitrosylation in late-onset AD [182]. Another protein that is aberrantly S-nitrosylated in AD is a neuronal cyclin-dependent kinase (CDK5) (Cys83, Cys157) [147]. The S-nitrosylation of CDK5 triggers Aβ-mediated dendritic spine loss and neuronal damage [145] (Table 1). This process involves the transnitrosylation of mitochondrial-fission protein DRP1 and the concomitant activation of CDK5 and NMDAR, which lead to the increased phosphorylation of glycogen synthase kinase-3β (GSK3β) and tau [145]. GSK3β contributes to the abnormal hyperphosphorylation of tau, and their S-nitrosylation suggests, as it was detected in a mouse model of AD, that it might potentially interfere with the neurofibrillary tangle’s formation [183,184,185]. The S-nitrosylation of the mitochondrial fission initiator, DRP1, has been extensively studied in AD pathogenesis [109,186,187]. More specifically, the incubation of rat cortical cell cultures and cortico-hippocampal splices with Aβ1-42, high glucose, or both led to S-nitrosylated DRP1-formation levels that are comparable to AD brains [146]. The S-nitrosylation of DRP1 results in excessive mitochondrial fragmentation, bioenergetic deficits and consequent synaptic loss [186,188] (Table 1). Remarkably, the blockade of DRP1 nitrosylation (by utilizing DRP1 (C644A) mutant) prevented both synaptic loss and neuronal cell death, suggesting that S-nitrosylated DRP1(Cys644) can be a promising therapeutic strategy for neuronal survival in AD [186].

A study utilizing quantitative proteomics, in which the authors assessed synaptosome S-nitrosylation in AD by utilizing proteins isolated from APP/PS1 model mice in comparison to wild-type and NOS2^−/−^ mice, as well as human control, mild cognitive impairment and Alzheimer’s disease brain tissues, is worthy of mention [148]. In the list of NOS2-dependent synaptosomal SNO proteins increased in aging and/or APP/PS1, genotype mice were included mitochondrial proteins, such as succinate dehydrogenase (ubiquinone) flavoprotein subunit, succinyl-CoA ligase (ADP-forming) subunit beta, acyl carrier protein, succinyl-CoA: 3-ketoacid coenzyme A transferase 1, NFU1 iron–sulfur cluster scaffold homolog and pyruvate carboxylase. The role of the S-nitrosylation modification of these mitochondrial proteins in AD and aging is still elusive. The uniqueness of this study was the inclusion of the inflammation-induced NOS2-dependent S-nitrosylation implication in synaptosomes during aging and PD and the provision of unique biomarkers for the inflammatory fingerprint in AD [148] (Figure 7).

## 5. Redox PTMs in Parkinson’s Disease

### 5.1. S-Glutathionylation in Parkinson’s Disease

#### 5.1.1. S-Glutathionylation/Glutathione in Parkinson’s Disease

Parkinson’s disease (PD) is a neurodegenerative disorder clinically characterized by resting tremor, rigidity, postural instability and bradykinesia [189]. During disease progression, affected individuals exhibit depression and cognitive deficits even earlier than motor symptoms [189]. The pathological hallmarks of PD are the gradual loss of dopaminergic neurons in the substantia nigra pars compacta (SN) region of the ventral mid brain as well as the presence of intracellular inclusions called Lewy bodies consisting mainly of aggregated α-synuclein (SNCA) [190]. The sporadic form of PD is correlated with multiple environmental factors, including the effects of neurotoxins such as 1-methyl-4-phenyl-1,2,3,6-tetrahydropyridine (MPTP), pesticides and herbicides such as rotenone and paraquat [191]. Familial PD cases constitute 10–15% of all PD cases, with mutations identified in more than 15 genes of Parkinsonism hereditary forms. The most renowned mutations in autosomal dominant PD include genes *SNCA* and *LRRK2* (enriched with leucine repeats kinase 2), while variants in *PARK2* (Parkin), *PINK1* (Pink1) and *PARK7* (DJ-1) genes mediate autosomal recessive and early onset PD [192]. Both oxidative stress and mitochondrial dysfunction represent cardinal events in neuronal loss processes during PD and have been strongly associated with both sporadic PD and familial PD cases [193]. Due to dopamine oxidation that produces ROS, dopaminergic neurons are particularly susceptible to oxidative stress. During PD, there is a selective inhibition of mitochondrial CI, leading to mitochondrial dysfunction and neuronal death. PD toxins such as rotenone and MPTP exert their function by the selective inhibition of CI [194]. In sporadic PD cases, selective CI inhibition might involve oxidative/nitrosative modifications of the different CI subunits [195].

Significant depletion of cellular glutathione has been observed in the SN of early PD patients and has been attributed to be the earliest known indicator of oxidative stress in pre-symptomatic PD, preceding both the decreased activity of CI and levels of dopamine [196]. Many studies have associated glutathione depletion in dopaminergic neurons with increased ROS levels and mitochondrial dysfunction due to the selective inhibition of CI activity most probably by thiol oxidation [197]. Consistent with increased oxidative stress in PD, decreased levels of reduced glutathione and increased levels of oxidized glutathione have been detected in SN postmortem PD tissues [198]. A decrease in erythrocyte glutathione concentration is correlated with more severe diseases based on the original Unified Parkinson’s Disease Rating Scale (UPDRS) scores [199], while lower serum glutathione is associated with worse Montreal Cognitive Assessment (MoCA) scores and Hoehn–Yahr staging, suggesting that glutathione levels might have utility as PD biomarkers [200]. The impact of oxidative stress and increased levels of oxidized glutathione in PD can be exemplified by the oxidative modifications of proteins shown to be mutated in familial PD cases and for which its oxidation might also interfere in disease pathogenesis. The Parkin protein, a RING-carrying protein with ubiquitin ligase activities involved in mitophagy regulation and immune-related functions, harbors cysteine residues that are susceptible to oxidative modifications with the treatment of Parkin with hydrogen peroxide to diminish its activity in vitro [201]. The role of a redox molecule has been attributed to highly oxidized Parkin proteins in the human brain, which decreases dopamine-metabolism-related stress. Specifically, it was discovered that Parkin can be S-glutathionylated in vitro at the conserved cysteine 59 and human-specific cysteine 95, and its modifications were reversed by GRX1 and GRX2 [201]. In mice treated with MPTP to assimilate PD, the inhibition of CI activity with a concomitant increase in glutaredoxin activity in the brain was observed, and this was followed by the damage inflicted upon selective dopaminergic neurons. Furthermore, the downregulation of glutaredoxin reversed the inhibition of CI activities [202,203]. It is known that the glutathionylation of Ndusf1 and Ndufv1 subunits in mitochondrial CI leads to a decrease in the activity of CI [35]. Despite the limitations of in vivo studies, it can be hypothesized based on current knowledge that glutaredoxin can reverse the glutathionylation of CI, preserving the activity of CI in sporadic PD. In a *Drosophila* PD model induced by the loss of the Parkin gene, ATP synthase β subunit glutathionylation levels decreased and were restored by GST omega (GstO). The GstO-mediated glutathionylation of the ATP synthase β subunit modified the activity of the mitochondrial F1F0-ATP synthase and had a protective role in restoring mitochondrial functions in the *Drosophila* PD model [153] (Figure 6).

#### 5.1.2. The Multifaceted Roles of GRX1 in Parkinson’s Disease 

Mutations in DJ1 are associated with early-onset autosomal recessive PD cases [204]. The DJ1 protein aims to provide neuroprotection by acting as a redox sensor [205]. Many roles have been attributed to DJ1, such as being a chaperone, protease, glyoxalase and transcriptional regulator for mitochondria protection against oxidative stress [154]. DJ1 triggers the expression of two mitochondrial uncoupling proteins (UCP4 and UCP5) that, in turn, decrease mitochondrial membrane potential and result in the inhibition of ROS production [206]. The stabilizing interaction between DJ1 and mitochondrial BCLXL protein modifies the activity of IP3R (inositol trisphosphate receptor), prevents cytochrome c release from mitochondria and inhibits the apoptotic cascade [207]. It has been shown that DJ1 translocates into mitochondria upon increased ROS production to exert its neuroprotective role [208]. DJ1 has been detected both in the mitochondrial matrix and in the intermembrane space, where it was colocalized with the NDUFA4 subunit of the mitochondrial CI even in the absence of stress [155]. Upon oxidative stress, DJ1 binding to CI subunits is enhanced. In human dopaminergic neurons, the knockout of the DJ1 gene results in mitochondrial depolarization, fragmentation and the accumulation of autophagy markers around the mitochondria [206]. Recent data demonstrated that DJ1 is glutathionylated in vitro and in vivo and that the glutathionylation levels modify its protein levels [156]. In addition, it has been identified that GRX1 regulates DJ1 levels in vivo [156]. This redox regulation of DJ1 sheds light on previous studies that exemplified the neuroprotective role of GRX1 overexpression in dopaminergic neurons in *C. elegans* [209]. Aside from GRX1, mitochondrial GRX2 might be a crucial regulator of mitochondrial homeostasis in PD. In particular, the overexpression of GRX2 protects against MPTP-induced cytotoxicity, whereas its downregulation disrupts Fe-S biogenesis and decreases mitochondrial CI activity in dopaminergic neurons similarly to the downregulation of GRX1 [210]. Despite the observation that DJ1 translocates into mitochondria to exert its neuroprotective role and the known interaction with GRX1, no data exist about the role of GRX2 in the oxidative regulation of DJ1 in mitochondria (Figure 6).

### 5.2. S-Nitrosylation in Parkinson’s Disease

Aberrantly increased ROS/RNS levels, as shown in the majority of sporadic cases in PD brains, phenocopied the effects of rare mutations in the gene encoding Parkin (*PARK2*) via the S-nitrosylation of Parkin-mediated mitochondrial dysfunction, protein misfolding and ubiquitin–proteasome system (UPS) impairment [150,211,212] (Figure 7). Many groups have shown that the S-nitrosylation of Parkin occurs in multiple cysteine residues at its RING and IBR domains and leads to the diminution of its E3 ligase activity and, consequently, defective UPS activity [150,151,161,213] (Table 1). Moreover, the S-nitrosylation of Parkin can irreversibly inactivate its enzymatic activity since S-nitrosylated Parkin promotes the formation of more stable oxidation products on the same cysteine residues, such as sulfinic acid (SO_2_H) and sulfonic acid (SO_3_H), via their reaction with ROS [212]. Intriguingly, the S-nitrosylation of Parkin transiently triggers its E3 ligase activity, followed by a diminution in activity [150,151]. The increase in E3 ligase activities that occur initially might reflect the neuroprotective aspect of S-nitrosylation signaling, whereas the following downregulation of its activity leads to UPS defects and the accumulation of misfolded proteins [150,151,152]. Under mitochondrial stress, full-length PINK1 stabilizes at the outer membrane of damaged mitochondria and phosphorylates a number of proteins, including Parkin and ubiquitin [214]. As a consequence, the PINK-mediated phosphorylation of ubiquitin triggers Parkin’s translocation to mitochondria and E3 ligase activity [215] and, subsequently, the polyubiquitinylation of outer mitochondrial membrane proteins and mitophagy [216]. Upon excessive NOS levels, the S-nitrosylation of PINK1(Cys568) inhibits its kinase activity, impairing the PINK1/Parkin-mediated mitophagy and thus leading to dopaminergic neuronal cell death [157] (Table 1). It is worthy of mention that the PINK1/Parkin pathway is impaired in many other neurodegenerative disorders, including AD and ALS [217]. Moreover, it was demonstrated that the S-nitrosylation of Parkin disrupts its ability to suppress mitochondrial fission protein DRP1, leading to DRP1 upregulation in vitro and in vivo [211]. In the same study, with respect to the susceptible neurons of MPTP-induced PD brains, DRP1 appears to be selectively increased. Furthermore, MPTP-induced NO-stress-triggered DRP1 phosphorylation (serine residue 616), which results in its recruitment to mitochondria. All these events manifest a death-prone milieu that contributes to dopaminergic neuronal loss. Interestingly, a nitric oxide synthase (NOS) inhibitor can suppress DRP1-mediated mitochondrial fragmentation in dopaminergic neurons, indicating that NO signaling is the linkage between MPP+ neurotoxicity and mitochondrial dynamics malfunction, which in part underlie the PD’s pathology [211]. An interesting recent discovery was the demonstration of the DJ1 requirement in the S-nitrosylation of Parkin under physiological conditions. In particular, the authors showed that DJ1 is required for the S-nitrosylation of Parkin and the interaction maintains mitochondrial homeostasis, whereas the inactivation of DJ1 leads to the denitrosylation of Parkin, contributing to PD pathology [218]. The novel discoveries about how NO signaling can contribute to PD pathology pave the way for promising therapeutic targets.

#### Mitochondria Biogenesis and Bioenergetics

Another pathway that could be a potential therapeutic avenue for PD is the nitrosative stress-induced MEF2C-PGC1a transcriptional network [158]. The authors demonstrated that in mitochondria-toxin-induced defects in A53T SNCA A9 human dopaminergic neurons, nitrosative/oxidative stress triggers the S-nitrosylation of transcriptional factor MEF2 (Cys39), which inhibits the MEF2C-PGC1a transcriptional network, resulting in mitochondrial dysfunction and apoptosis (Table 1). These data provide insight into the NO-signaling interferences within the gene-environmental interaction that might contribute to PD pathogenesis. Many studies assessed the NO-mediated nitrosylation and nitration of CI subunits in various models of PD and demonstrated that CI inhibition is induced by both the NO-mediated nitrosylation and nitration of CI subunits [149] (Table 1). Recently, researchers demonstrated that the administration of inorganic nitrite corrects mitochondria dysfunctions in PD patients’ fibroblasts, mitigating PD pathology [219]. Mechanistically, the protective effect of nitrite is mediated by both the S-nitrosylation of CI and the downstream activation of the antioxidant NRF2 pathway [219]. Currently, nitrite is being tested in clinical trials for cardiovascular disorders [220]; therefore, it could be easily repositioned for PD treatments. In addition, the S-nitrosylation of mitochondrial chaperone PHB has neuroprotective roles against oxygen and glucose deprivation stress [221], although its expression is reduced in PD brains [160]. Moreover, in PD patients’ brain tissues and PD models, the S-nitrosylation of peroxiredoxin (PRDX2) (Cys51, Cys172) diminishes peroxidase activities, causing hydrogen peroxide to accumulate and exacerbates oxidative stress [159] (Table 1). 

Concerning SNCA (which has no cysteine residues), the primary NO-induced modification is the nitration of its turbine residues in PD [222]. SCNA nitration (Tyr39) can potentiate SNCA oligomer formation [162] (Table 1). Indeed, PD patients’ serum-nitrated SNCA levels correlate with worsened PD outcomes [223], suggesting its usage as a potential biomarker. Interestingly, a recent study demonstrated that extracellular SNCA oligomers induce oxidative/nitrosative stress that led to the S-nitrosylation of Parkin. In turn, this modification results in the autoubiquitination of Parkin and degradation, contributing to significant cell death [161]. These data suggest that the extracellular SNCA-mediated nitrosylation of Parkin might contribute to the propagation of neurodegeneration in PD pathology.

## 6. Redox PTMs in Amyotrophic Lateral Sclerosis

### 6.1. S-Glutathionylation in Amyotrophic Lateral Sclerosis

Amyotrophic lateral sclerosis (ALS) is a severe late-onset neurodegenerative disorder that affects primarily the upper and lower motor neurons. ALS patients develop progressive paresis and skeletal muscle atrophy ultimately leading to quadriplegia and fatal respiratory failure. Almost 90% of the patients fall under sporadic ALS cases (sALS) with no affected first-degree relatives, whereas approximately 10% have a familial predisposition (fALS) [224]. In the last thirty years, more than 40 mutations have been identified in ALS patients, with the second most frequent involving mutations in the *SOD1* gene, which encodes for Zn, Cu superoxide dismutase [225]. Due to the increased variabilities in clinical phenotype with intra- and inter-familial differences, ALS has a highly heterogeneous clinical phenotype [226]. The longstanding zeitgeist in SOD1 protein conformational changes and function stems from the over 100 different mutations in *SOD1* gene, which accounts for the fALS cases as well as the presence of amyloid plaques consisting of different SOD1 species [227]. In the past, the abnormal accumulation of SOD1 fibrils was suggested to be a toxic factor in disease manifestation; the currently prevailing hypothesis is that the SOD1 trimer is toxic to neurons in ALS [228], whereas the SOD1 fibrils rather constitute a protective mechanism for neurons [229]. However, the unique characteristics of SOD1 and the complexity of its conformation states render unclear whether ALS-related SOD1 mutations are causative or modify the disease.

#### 6.1.1. SOD1 S-Glutathionylation

The human SOD1 protein, an essential antioxidant enzyme, exists within neurons as a mixture of apoSOD1 (monomeric and dimeric metal-free) and holoSOD1 (partially or fully metallated dimeric) [227]. The maturation of SOD1 from apoSOD1 to a holoSOD1 dimer depends upon key post-translational modifications; the binding of Zn and Cu to each monomer, intramonomeric disulfide formation and homodimerization. The unique characteristic of SOD1 conformation is this stable disulfide bond between Cys57 and Cys146 conserved in each monomer, and it is the only known example of this type of bond to exist as an oxidized state in a mature and functional protein within a reducing environment of the cytosol [227]. Alterations in those modifications that stabilize SOD1 proteins or atypical modifications on specific residues are associated with mutant or wild-type SOD1 misfolding. In particular, the oxidation of Trp32 and glutathionylation of C111 both trigger SOD1 dimer disassembly and aggregation [163,230]. On the one hand, Trp32 is found at the core segment of SOD1 (residues 28–38), playing a role in SOD1 oligomerization, and its oxidation is shown to be involved in triggering SOD1 self-assembly [230]. On the other hand, Cys111 is found within the Greek key loop, playing a role in stabilizing the dimer’s interface within the complex hydrogen-bonding network. For over a decade, the oxidation of Cys111 via glutathionylation (S-SG) in SOD1 proteins was studied, and it was suggested to be the factor that destabilizes the SOD1 dimer, increases the probability of the monomer’s unfolding and consequent aggregation and results in the SOD1’s loss of activity and cell death [163] (Figure 6) (Table 1). Conversely, the phosphorylation of T2 might inhibit dimer disassembly by stabilizing the dimer’s interface, suggesting that phosphorylation in T2 is a compensating mechanism for the glutathionylation of Cys111 in mutated forms of SOD1 under oxidative stress [231]. Interestingly, the oxidation of wild-type SOD1 adopts a similar conformation relative to mutant SOD1 conformation, which could be pathogenic in sporadic ALS, which constitutes 90% of ALS cases [232]. Additionally, SOD1 misfolding and aggregation under an oxidized environment are not only evident in familial and sporadic ALS but also in sporadic Parkinson’s and Alzheimer’s disease, suggesting that this pathological feature is shared among neurodegenerative diseases [227]. 

#### 6.1.2. SOD1 Role in Mitochondria

SOD1 is a superoxide oxidase and reductase, which breaks down superoxide to produce molecular oxygen and, in turn, reduces superoxide into hydrogen peroxide, which can be further metabolized by catalase into water and oxygen. Human SOD1 is within the group of frontline antioxidant mechanisms against oxidative stress in all eukaryotic cells since the evolution of aerobic respiration by mitochondria increased the mitochondrial and cellular production of superoxide [227]. SOD1 is highly abundant in the cytosol, distinguishing it by Mn superoxide dismutase (SOD2) solely localized in mitochondria and extracellular Cu, Zn superoxide dismutase (SOD3) anchored in the extracellular matrix [233,234]. Despite its cytosolic localization, the SOD1 pro-survival effect seems to be most critical within the inter membrane space (IMS) of mitochondria since an IMS-targeted SOD1 adequately protects cells from oxidative stress [235]. Additionally, in SOD1^−/−^ axons, mitochondria densities decreased and might contribute to distal motor axons deficits due to their great demand for energy inputs. Immature SOD1 and CCS (copper chaperone for superoxide dismutase), which also plays a role in SOD1 maturation by inserting Cu, are imported into the mitochondrial intermembrane space by the translocase of the outer mitochondrial membrane, where they are both trapped by metal insertion and disulfide bond formation mediated by MIA40/ERV1’s disulfide relay system [235,236]. Upon the sudden increase in mitochondrial oxidative stress, CCS-dependent SOD1 maturation within IMS traps SOD1 within the compartment to mitigate acute oxidative insult, and this augments SOD2-mediated superoxide clearance within the mitochondrial matrix [235,236]. 

Multifaceted molecular pathways might correlate SOD1 self-assembly and neurotoxicity [227]. The role of SOD1 in mitochondria has been extensively reviewed [237]. Regarding the impact of SOD1 misfolding in cellular homeostasis via mitochondria-associated pathways, we summarize below the current hypotheses for the misfolded SOD1 species’ involvement in mitochondria homeostasis in ALS, which could be interesting starting points for investigating SOD1 functions in the mitochondria of other SOD1-related neurodegenerative disorders (Figure 6):Misfolded SOD1 events in the outer mitochondrial membrane: One hypothesis is that SOD1 small misfolded species trigger mitochondrial cytochrome c release and caspase-dependent-programmed cell death. In both in vitro (G93A, G85R) and mutant SOD1 murine models (G93A, G37R), the accumulation of misfolded mutant SOD1 oligomers on the outer mitochondrial membrane has been proposed to trigger apoptosis [238]. This misfolded SOD1 localization has been shown to be highly BCL2-dependent in cell cultures, mutant SOD1 murine models and SOD1-linked familial ALS patients [239,240]. BCL2 halts the release of pro-apoptotic factors from mitochondria, such as cytochrome c, preventing caspase activation and apoptosis under physiological conditions. Pedrini et al. showed that misfolded SOD1 binding triggers conformational changes in BCL2, resulting in the exposure of its toxic BH3 domain and, thus, triggering cytochrome c release and eventually apoptosis [239]. Importantly, GRX1 and GRX2 can reduce disulfides to protein thiols that prevent mutated SOD1 aggregation and rescue mitochondria function while preventing neuronal cell apoptosis [241]. In addition, it has been shown that misfolded SOD1/BCL2 interactions decrease the mitochondrial membrane’s permeability relative to ADP by direct inhibition of the outer mitochondrial porin voltage-dependent anion channel 1 (VDAC1), which regulates mitochondrial ATP production and export [242]. This toxic conformation of BCL2 triggered by misfolded SOD1 can lead to bioenergetic defects, increased levels of ROS and calcium homeostasis deregulation, as has been shown in the motor neurons of pre-symptomatic G93A mice [239,240]. Interestingly, oxidized wild-type SOD1 recapitulates the same toxic behavior, pinpointing the possible common pathogenic mechanism and, thus, potential therapeutic targets between mutated SOD1-related fALS and sporadic disorders exhibiting oxidized wild-type SOD1, including sporadic ALS and Alzheimer’s and Parkinson’s diseases [163].Misfolded SOD1 events inside mitochondria: As mentioned above, apoSOD1 and CCS can translocate within the IMS, mitigating superoxide emission and triggering SOD2-mediated superoxide detoxification within mitochondria. However, if apoSOD1 becomes misfolded inside the mitochondria, a substantial accumulation of SOD1 aggregates can happen within this sub-compartment, which is associated with electron transport chain defects and increased ROS production. In this case, it has been suggested that the reduced electron transport chain function is attributed to the preference for the delivery of Cu to SOD1 at the expanse of mitochondrial cytochrome c oxidase [227]. Banks et al. were the first to identify that acylation on SOD1 modifies its capability to suppress mitochondrial respiration [243]. The authors showed that increased levels of sirtuin activity deacylated SOD1, which in turn, activates SOD1 respiration-suppression activity at the CI of ETC. Sirtuin activities are linked to NAD+ levels, which are linked to the overall metabolic cellular status, suggesting that the acylation of SOD1 might act as a sensor link between nutrient metabolism and the SOD1-mediated suppression of respiration [243]. While this identification may add an additional role to SOD1, the mechanistic pathway with which this SOD1-mediated suppression of respiration correlates to SOD1-mediated cell survival is still uncertain.Misfolded SOD1 and mitochondrial integrity: The impact of *SOD1* gene mutations on mitochondrial integrity has been uncovered by utilizing human iPSC-derived spinal cord motor neurons (MNs) of three quintessential *SOD1* gene mutations [244]. It has been shown that mitochondria integrity defects precede DNA damage in neurons from patients with SOD1 mutations, suggesting that a mitochondrial homeostasis deregulation is an upstream event in SOD1-related ALS. In particular, MNs expressing SOD1 R115G and D90A showed elongated mitochondria, highly reduced membrane potential and intracellular ATP levels and a higher fraction of moving mitochondria. In contrast, MNs expressing SOD1 A4V, while they did not show those morphological/motility alterations, had mitochondrial membrane potentials that were highly reduced [244]. The observation of mitochondria elongation in SOD1 D90A and R115G could be explained by protective mitofusion or hyperfusion that can act as a protective mechanism for mitochondria dysfunction against mitochondrial fission and macro-autophagy [244]. The misexpression of mitochondrial dynamics genes has been associated with mutant SOD1 in ALS [245]. Excessive mitochondrial fission and increased mitochondrial fragmentation have been reported in both ALS-patient-derived fibroblasts and the motor neuron cultures of multiple familial forms of mutated SOD1 ALS expression [246]. Interestingly, the inhibition of DRP1/FIS1 by a selective peptide inhibitor, P110, led to a significant improvement of mitochondria structure and function in mice expressing G93A SOD1 mutations [246].

### 6.2. S-Nitrosylation in Amyotrophic Lateral Sclerosis

In addition to increased protein S-nitrosylation, the decreased S-nitrosylation of specific proteins may also lead to disease pathogenesis in the brain [247]. The disruption of S-nitrosothiol homeostasis may result either from defective NOS activities or from impairments in the denitrosylases that remove NO groups. One of the hypotheses about ALS manifestation is that the increased denitrosylase activity of mutant SOD1 causes a depletion of intracellular S-nitrosothiols, disrupting the cellular or subcellular function of proteins that are regulated by S-nitrosylation and ultimately contributing to motor neuron death [247] (Figure 7). Mutant-SOD1 increased denitrosylation is particularly prominent in mitochondria [247], for which its dysfunction plays an important role in familial and sporadic ALS pathogenesis. Despite the observation that the excessive denitrosylation activities of mutant SOD1 might manifest pathological roles in ALS, the lower denitrosylase activity of WT SOD1 might play a role in S-nitrosothiols homeostasis. While WT SOD1 denitrosylase activities are not that efficient, the activities might denitrosylate proteins in privileged sites where more efficient denitrosylases, such as S-nitrosoglutathione reductase, are inaccessible [248]. Indeed, there is no evidence thus far showing that the GSNO reductase resides in mitochondria; therefore, WT SOD1 might regulate mitochondrial S-nitrosothiols homeostasis. The expression of mutant SOD1 in mitochondria, but not in the nucleus or endoplasmic reticulum, leads to neuronal cell death [249]. Additionally, mutant, but not WT SOD1 forms, aggregates in spinal cord mitochondria that trap BCL2, hampering its anti-apoptotic efficiency [250,251]. In one particular study, it was shown that SNO levels are higher in the spinal cord mitochondria of transgenic mice expressing a copper-deficient compared with a copper-repleted SOD1 mutant [247]. This observation supports the hypothesis that mitochondrial vacuolization manifests only in transgenic mice expressing mutant SOD1 that binds copper efficiently to deplete mitochondria S-nitrosothiol levels [252]. S-nitrosothiol donor compounds could be a novel therapeutic intervention in ALS and other neurodegenerative disorders with S-nitrosothiol depletion, as the repletion of intracellular S-nitrosothiol levels with S-nitrosothiol donor compounds rescued mutant-SOD1-induced cell death [247]. Aside from disrupting cell signaling, the increased mutant SOD1 denitrosylation of thiols might lead to aberrant disulfide bond formation, which in turn contributes to protein aggregation in mutant SOD1 cell cultures [253]. Concerning protein misfolding in ALS, another recent observation demonstrated that the S-nitrosylation of PDI (protein disulfide isomerase) inhibits its activity, triggers mutant SOD1 aggregation and increases neuronal cell death [142]. Therefore, the denitrosylation of PDI can be stratified as a therapeutic intervention.

Concerning the NOS expression and implication in ALS, it has been shown that increased NO levels in mutant SOD1 mice correlate with inducible NOS (iNOS) rather than neuronal NOS (nNOS) [254]. Interestingly, mitochondria produce NO by a mitochondrial form of NOS (mtNOS) with similar substrates and cofactor requirements relative to constitutive NOS, although mtNOS has been shown to cross-react immunologically with antibodies against iNOS [255,256]. In line with this, in ALS mice, iNOS is catalytically active in spinal cord mitochondria during the time frame that iNOS is highly expressed in motor neurons but not microglia [257]. This finding implies an iNOS-induced NO-mediated mechanism for the mitochondriopathy observed in the motor neurons of mutant SOD1 mice. Concerning protein misfolding in ALS, it has been demonstrated that the iNOS-mediated S-nitrosylation of PDI inhibits its activity, triggers mutant SOD1 aggregation and increases neuronal cell death [142] (Table 1). There is extensive accumulating evidence for increased S-nitrosylated PDI levels in the brains of sporadic ALS, PD and AD [258]. Thus, the denitrosylation of PDI can be utilized as a common therapeutic approach for many sporadic cases.

## 7. Redox PTMs in Friedreich’s Ataxia

### 7.1. S-Glutathionylation in Friedreich’s Ataxia

Friedreich’s Ataxia (FRDA) is an autosomal recessive neurodegenerative disorder, which prevails among the inherited forms of ataxia, as it affects 1 in 50,000 individuals [259]. The underlying cause of FRDA is frataxin (FXN) deficiencies resulting from either a biallelic GAA trinucleotide repetition (ranging from 200 to 1700 GAA), which constitutes 95% of patients (homozygous) or a point mutation in one *FXN* allele, which is paired with an expanded allele and represents 5% of patients (heterozygous) [260]. Patients with FRDA develop progressive cerebellar and sensory ataxia as well as lower limb pyramidal weakness and dysarthria [261]. Those affected by FRDA often manifest diabetes mellitus, cardiomyopathy and skeletal abnormalities [262]. Frataxin is a mitochondrial protein that was identified in the 1990s, but its functional role is still controversial [263]. The most prevailing role of frataxin is the one related to iron metabolism in terms of an iron-binding evolutionarily conserved protein, which resembles ferritin [264], since its ability to form oligomers with high molecular weight by interacting with iron has been proposed [265]. Amongst the most credited frataxin’s functions is its involvement in iron–sulfur cluster biosynthesis (ISC), in which frataxin has been proposed to interact with the complex formed by the iron–sulfur cluster assembly enzyme ISCU and cysteine desulferase NSF1 [266], even if it is not yet elucidated whether frataxin acts as an allosteric regulator [267] or iron donor [268].

The diminution of mitochondrial protein frataxin leads to increased mitochondrial iron accumulation, decreased ISC synthesis and impaired mitochondrial function [269]. In the last twenty years, a significant amount of research in FRDA pinpoints the contribution of oxidative stress, ISC-impaired biogenesis and glutathione metabolism in disease manifestation, although the exact mechanism with which the three aforementioned systems interact and contribute to disease progression or exacerbation is not yet clear. The main phenotype in FRDA is iron accumulation inside the mitochondria accompanied by compromised ETC CI, II, III and ACN activities, ultimately leading to a Fenton-mediated overproduction of superoxide and hydroxyl radicals [270,271] (Figure 6). Oxidative-stress-induced damages such as lipid peroxidation and ROS overload have been observed in FRDA mouse models, yeast and FRDA patients’ fibroblasts. Oxidative stress and mitochondrial respiratory complex interplay have been extensively studied in multiple FRDA models with contradicting results. While data from yeast and mouse models provide evidence of disrupted ETC complexes, defective mitochondrial respiration was not a result of the global impairment of ETC complexes but rather a result of the inactivation of KGDH and ACN [55]. Consequently, the redox regulation of KGDH via glutathionylation limits the production of NADH and electron flow in the respiratory chain in yeast-deficient frataxin cells and FRDA patients’ fibroblasts [55]. Decreased levels of total glutathione have been described in yeast and FRDA mice in vivo and patients’ fibroblasts with a concomitant abnormal increase in mitochondrial and cytosolic proteins glutathionylation [55]. In particular, aside from the observed glutathionylation of KGDH and CIII and CIV, actin glutathionylation prevails in many studies in which iron-induced oxidative stress correlated with prominent cytoskeletal abnormalities in FRDA [272]. Of particular interest is the downregulation of the transcription factor, nuclear factor erythroid 2-related factor 2 (NRF2), which seems to have multifaceted outcomes. NRF2 is the primary antioxidant mechanism in neurons and its downregulation has been observed in FRDA [273]. The compilation of impaired glutathione metabolism and mitochondria function as well as defected ISC biosynthesis with a concomitant decrease in protein levels of GPX4 and NRF2 mounted the role of ferroptosis in FRDA as a pathogenic mechanism leading to cell death [274].

### 7.2. S-Nitrosylation in Friedrich’s Ataxia

Many studies extensively analyzed the interaction between frataxin, iron homeostasis and nitric oxide signaling, mainly in *Arabidopsis* roots and yeast [275,276]. In particular, it has been shown that diminutions in frataxin levels cause iron accumulation, which in turn produces hydroxyl radicals via Fenton’s reaction [275]. Due to the abnormal increase in ROS and iron contents, there is a concomitant increase in NO production that exerts antioxidant properties in a dualistic manner: directly by scavenging peroxide and indirectly by the NO-mediated triggering of ferritin proteins, which diminishes free iron levels within organelles, thus protecting against oxidative stress and cell toxicity [275]. In FRDA patients, frataxin deficiency led to the decreased activity of mitochondrial CI, II, III and ACN [269]. Both CI and ACN have been found to be targets of S-nitrosylation, which inhibits their activity [164]. Whether the observed increased NO production mediates the S-nitrosylation of mitochondria iron–sulfur-containing proteins and in turn diminishes their activity upon frataxin deficiency and increased mitochondrial iron accumulation has not yet been elucidated. Interestingly, an increasing amount of evidence suggests that iron homeostasis regulatory proteins are modified upon nitrosative stress by S-nitrosylation, and this modification triggers signaling cascades that cause neuronal cytotoxicity [33]. In particular, the S-nitrosylation of iron regulatory protein 2 (IRP2) (Cys178) can result in the malfunction of iron homeostasis via the UPS-dependent degradation of IRP2 that results in the increased accumulation of iron inside iron storage protein ferritin, thus implicating the S-nitrosylation of IRP2 protein in nitrosative-stress-mediated neurotoxicity [165] (Figure 7) (Table 1). Accordingly, *Irp2* knockout mice manifest impaired iron metabolism, movement disorders and neurodegeneration [166]. These results suggest that the S-nitrosylation of IRP2 impairs iron metabolism and leads to nitrosative-mediated neurotoxicity.

In a recent study, it has been demonstrated that the levels of SOD1 and mitochondrial SOD2 decreased in FRDA patients’ fibroblasts. These observations were also corroborated in frataxin-deficient mice [277]. When levels of SOD enzymes diminish, the toxic superoxide anion is free to react and form highly reactive anions inside the cell, such as peroxynitrite, by reacting with nitric oxide. The following formation of peroxynitrite can trigger caspase-dependent apoptotic cell death [278]. Remarkably, upon the addition of bone-marrow-derived mesenchymal stem cell (MSCs)-conditioned medium in FRDA fibroblasts upon nitrite oxide stress, there is an increase in frataxin and SOD levels, followed by increased cell survival. These results suggest that the MSC-conditioned medium possesses the ability to modulate nitrosative-stress-mediated deficient signaling by inducing antioxidant defenses in FRDA cell cultures [279].

## 8. Concluding Remarks

The trigonal interaction between ROS/RNS, mitochondria dysfunction and proteinopathy has been extensively studied in multiple neurodegenerative disorders, including AD, PD, ALS and FRDA. Intriguingly, which of these cardinal pathological characteristics constitutes the cornerstone of neurodegenerative disorders is still uncertain, as all three manifestations can both trigger and exacerbate disease statuses in sporadic and familial cases (Figure 8).

While researchers are still trying to identify the cause of neurodegeneration during aging and genetic disorders, the only constant factor present in all cases is oxidative/nitrosative stress. Due to this knowledge, there was a long-lasting endeavor to treat neurodegeneration with antioxidant molecules, but the promising results in mouse models and pre-clinical studies could not be reproduced and, in turn, failed in clinical trials [34]. Consequently, research has shifted towards the elucidation of the oxidative/nitrosative stress’s imprint in cellular architecture with the ultimate goal being to decipher the mechanistic pathways with which redox-status alterations trigger or exacerbate pathological conditions [25,27]. In the last few decades, redox post-translational modifications, such as S-glutathionylation and S-nitrosylation, have gained increased attention due to the overwhelming evidence that implicates them in cell-signaling processes. Importantly, their role in neurodegenerative disorders is just starting to be elucidated [25,27]. In AD, PD, ALS and FRDA, both S-glutathionylation and S-nitrosylation appear to exacerbate the disease status by modulating cardinal pathways, particularly in the mitochondria, and in turn phenocopied the mutation-related familial cases of those diseases, as have been extensively described in this review (Figure 6 and Figure 7) (Table 1). At the same time, their exact role in even mitochondria homeostasis is still elusive, and this motivates further investigations with respect to their mechanisms of function and impacts on physiology.

Given the fact that there is still no cure for most neurodegenerative disorders, the stratification of redox PTMs as an alternative therapeutic approach seems to be a very appealing strategy. The modification of the redox-PTMs of proteins with a pivotal role in neurological disorders by utilizing small molecules that could restore the normal function of the affected proteins has already been proposed [17,280,281]. For instance, deprenyl/selegiline and its derivatives (CGP3466B and TCH346) have already shown positive results in multiple neurodegenerative disorders, such as AD, PD, ALS, muscular atrophy, traumatic brain injury and ischemia [282,283,284,285,286,287]. The mode of action of these compounds is based on their ability to bind to GAPDH, preventing it from S-nitrosylation under nitrosative/oxidative stresses and, thus, blocking the apoptotic cascade that is triggered by S-nitrosylated GAPDH–SIAH1 interactions. These observations pinpoint the significance of further elucidating how redox-PTMs function and their application in reversing neurodegenerative processes. 

Our knowledge so far exemplifies the roles of S-glutathionylation and S-nitrosylation in neurodegeneration. Concerning the information gathered from old and novel S-glutathionylation targets in AD, PD, ALS and FRDA (Figure 6), it can be deduced that S-glutathionylation can be stratified in a dualistic manner. On the one hand, it can be a promising indication of both aging status and disease progression; on the other hand, when utilized as a biomarker, S-glutathionylation implications in fusion, mitochondria bioenergetics and protein misfolding could be ideal candidates for therapeutic intervention for many neurodegenerative disorders. In the case of S-nitrosylated proteins, which in many cases phenocopied the effect of mutated genes in the familial cases of neurodegenerative disorders (Figure 7), they can also be utilized as potential targets for preventing excessive mitochondrial fission and protein misfolding. The main question, which still has not been discussed in neurodegenerative disorders, is the crosstalk between S-glutathionylation and S-nitrosylation in disease manifestation. Could it be that the concurrent modulation of S-glutathionylated and S-nitrosylated targets in the same disease fine-tunes impaired cellular homeostasis more efficiently? Another question, particularly for genetically predisposed neurological disorders, could be whether the combinatorial therapy that corrects the mutated genes and at the same time modifies the redox proteome holds the key to a powerful treatment for neurodegeneration.

Altogether, further developments in redox-based proteomic approaches not only qualify for mapping the redox proteome homeostasis and deregulation in different redox-influenced disorders but also qualify for the designation of targeted therapeutic strategies.

## Figures and Tables

**Figure 1 ijms-23-15849-f001:**
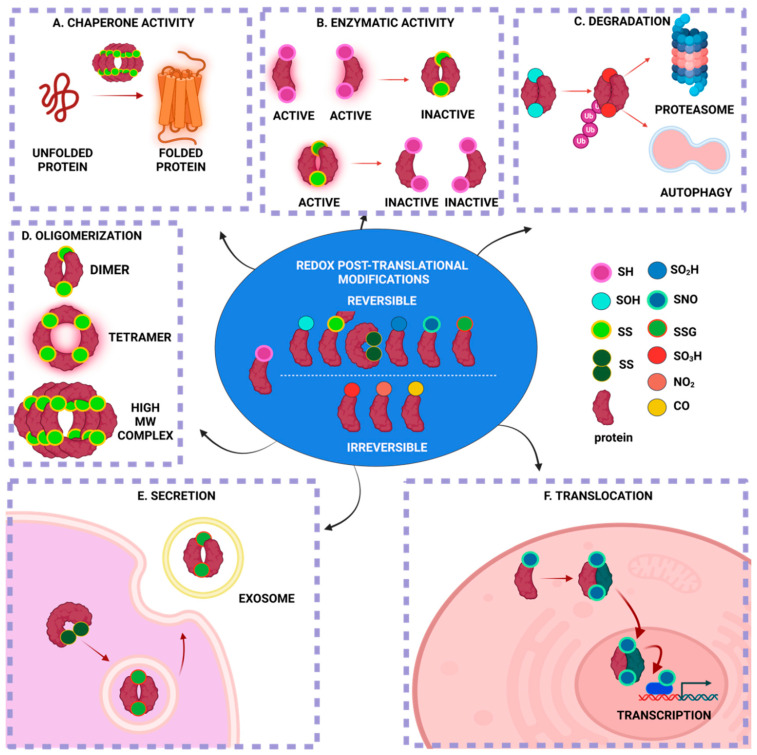
Redox post-translational modifications effects on proteins structure, function, location and turnover. ROS/RNS can modify amino acid residues of a protein by S-sulfenylation (–SOH), intermolecular and intramolecular disulfide bonds (–SS–), S-sulfinylation (–SO_2_H), S-nitrosylation (–SNO), S-glutathionylation (–SSG), S-sulfonylation (–SO_3_H), nitration (–NO_2_) and carbonylation (–CO: carbonyl groups). Based on its redox PTMs, a protein can act as a chaperone (**A**), change the conformation that activates or deactivates its enzymatic function (**B**), irreversibly oxidize it and trigger ubiquitin-mediated proteasomal degradation (**C**), modulate its conformation and oligomerization (**D**), regulate its secretion in extracellular space (**E**) and trigger its translocation into sub-compartments, thus acting as a signaling molecule (**F**). Created with BioRender.com.

**Figure 2 ijms-23-15849-f002:**
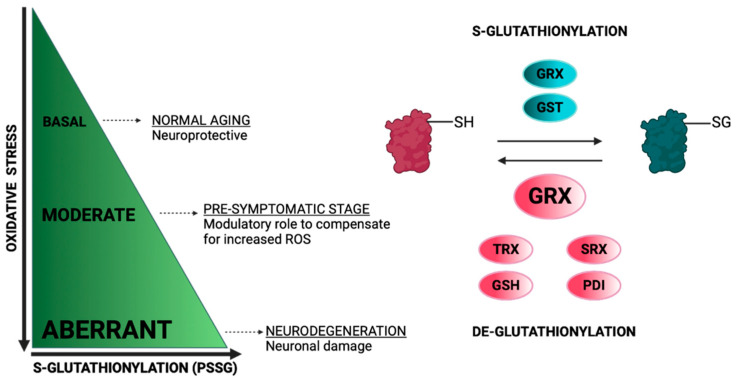
General mechanisms of S-Glutathionylation and Deglutathionylation. Under oxidative stress, proteins can be S-glutathionylated non-enzymatically or enzymatically via GRX and GST. The reverse reaction, deglutathionylation, takes place non-enzymatically, upon increased levels of GSH to GSSG, or enzymatically, not only via GRX primarily but also TRX, SRX and PDI. During aging, a subset of proteins is S-glutathionylated, promoting neuroprotection. Upon increased levels of ROS, potentially at early stages of neurodegeneration, increased levels of S-glutathionylation protect proteins from irreversible oxidation and degradation while simultaneously modifying cellular homeostatic mechanisms to compensate for increased ROS. Aberrant S-glutathionylation in neurodegenerative disorders exacerbates disease pathology, ultimately leading to neuronal damage. Created with BioRender.com.

**Figure 3 ijms-23-15849-f003:**
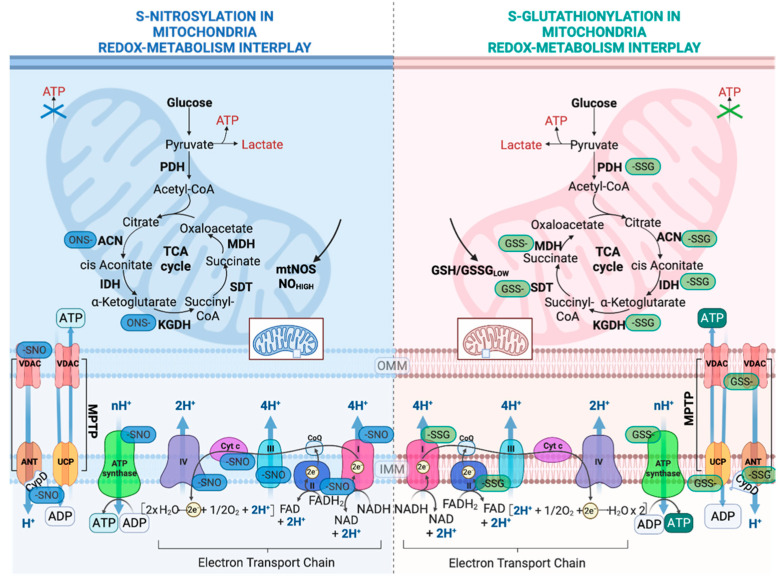
S-Nitrosylation and S-Glutathionylation roles in mitochondria redox-metabolism interplay. The targets of S-glutathionylation in the TCA cycle are PDH, ACN, IDH, KGDH, SDT and MDH. The S-glutathionylation of TCA cycle enzymes protects them from irreversible oxidation until the redox status is impaired and inhibits ROS production and electrons transfer to the ETC to further minimize the ROS levels under oxidative stress. Complex I and V inhibitions by S-glutathionylation share similar purposes with the S-glutathionylation of TCA cycle enzymes and ultimately minimize ROS production. Only Complex II S-glutathionylation has been shown to be integral for its constitutive activity. Under excessive ROS, the glutathionylation of ANT and VDAC prevents the formation of MPTP in inhibiting apoptosis, whereas UCP glutathionylation increases ROS formations. The targets of S-nitrosylation in the TCA cycle are ACN, KGDH and SDH, which in turn inhibits their function to minimize ROS. All mitochondria Complexes I-V can be nitrosylated under increased oxidative/nitrosative stress both to minimize ROS production and to scavenge NO. Under basal NO levels, the S-nitrosylation of VDAC and MPTP constituents inhibits their functions in providing protection from apoptosis, whereas under excessive NO levels, aberrant S-nitrosylation activates VDAC-mediated cytochrome c release and apoptosis or/and CYPD S-nitrosylation, the activation of MPTP, the ablation of ATP production and necrosis (OMM: outer mitochondrial membrane; IMM: inner mitochondrial membrane). Created with BioRender.com.

**Figure 4 ijms-23-15849-f004:**
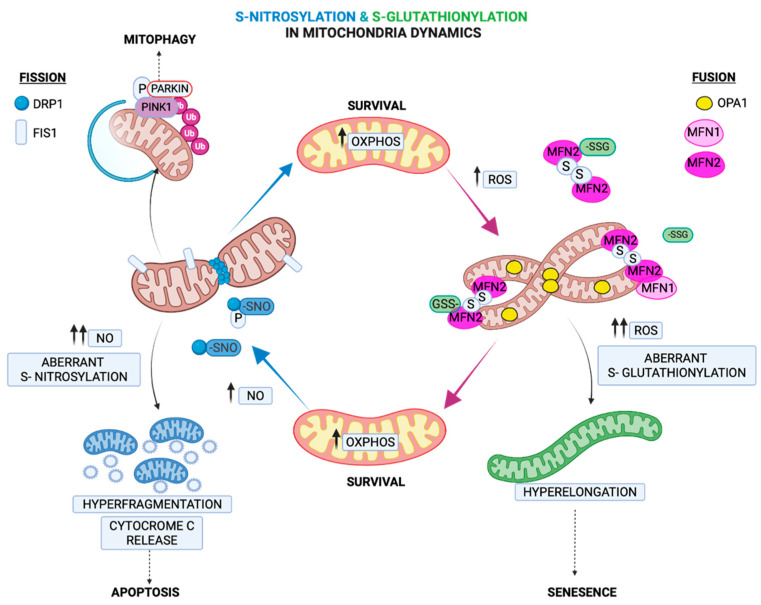
S-Glutathionylation and S-Nitrosylation orchestrate mitochondrial dynamics. Under acute stress, the S-glutathionylation of MFN2 and, in turn, the oligomerization of OPA1 trigger hyperfusion as an adaptive mechanism, whereas the excessive hyper-elongation of mitochondria under aberrant S-glutathionylation leads to cell senescence. The NO-mediated S-nitrosylation of DRP1 triggers its activation and phosphorylation, which in turn activates the fission machinery to eliminate damaged mitochondria. Under excessive NO production, the hyper-fragmentation of mitochondria leads to cytochrome c release and apoptosis. Created with BioRender.com.

**Figure 5 ijms-23-15849-f005:**
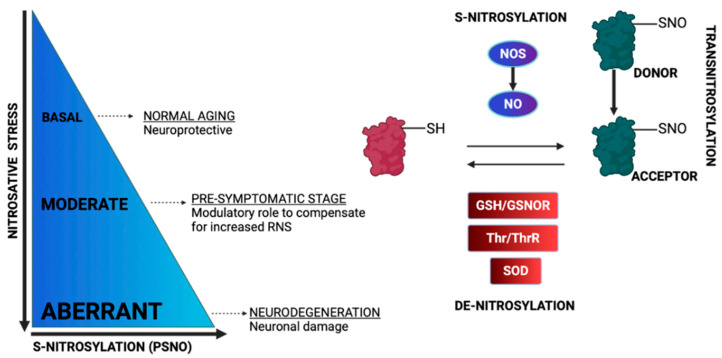
Proposed mechanisms for S-Nitrosylation and Denitrosylation. While NO-mediated protein S-nitrosylation is generally a non-enzymatic reaction, mediated by RNS and ROS species and NOS-mediated NO production, denitrosylation can be both enzymatic and non-enzymatic. The major denitrosylase systems in cells are the Thr/ThrR and glutathione GSH/GSNOR. Moreover, SOD can have denitrosylase functions. The transnitrosylation of a protein can be directly performed by an already S-nitrosylated protein in close proximity. During aging, a subset of proteins is S-nitrosylated, promoting cell signaling cascades that aim for neuroprotection. Upon increased levels of RNS, potentially at early stages of neurodegeneration, increased levels of S-nitrosylation modulate multiple cellular processes and trigger the adjustment of signaling cascades in highly nitrosative environments. Aberrant S-nitrosylation in neurodegenerative disorders exacerbates disease pathology, ultimately leading to neuronal damage. Created with BioRender.com.

**Figure 6 ijms-23-15849-f006:**
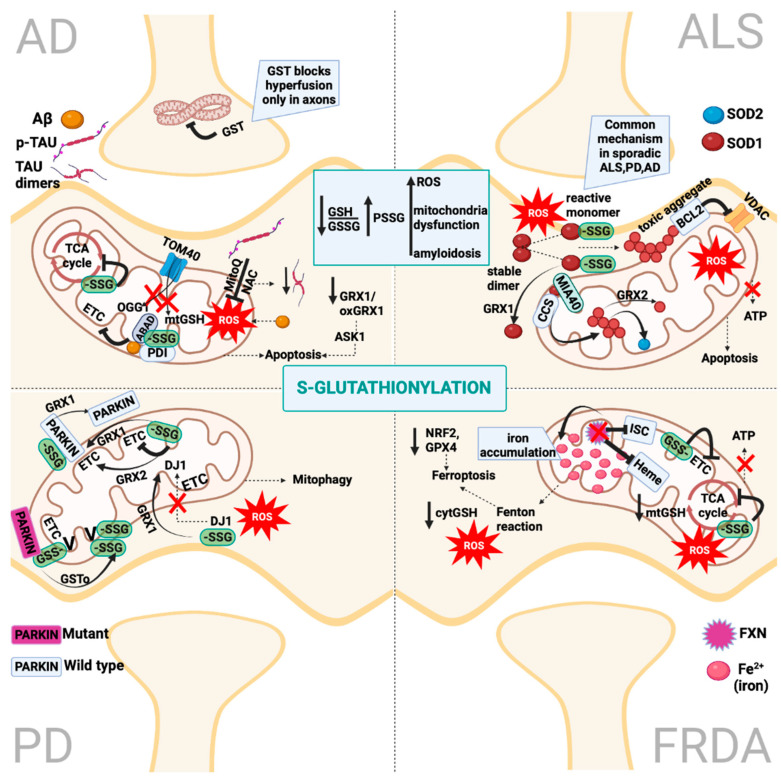
S-Glutathionylation impairs mitochondria homeostasis in AD, PD, ALS and FRDA. In AD, aberrant S-glutathionylation due to decreased levels of cytosolic and mitochondrial GSH inhibits the TCA cycle and ETC, impairing mitochondria respiration. Polymorphisms in the *TOMM40* gene inhibit the import of GSH and OGG1 inside mitochondria. SOD1 has been found inside the mitochondria matrix and is colocalized with S-glutathionylated ABAD and PDI, interfering with ETC functions. The addition of MitoQ/NAC decreases the ROS production associated with phosphorylated Tau in AD models but decreases the amount of dimeric Tau. Both reduced levels of GRX1 and oxidized GRX1 cause ASK1-mediated apoptosis in AD. In PD, the S-glutathionylation of ETC inhibits its activity, which is reversed by GRX1 and GRX2. The S-glutathionylation of DJ1 inhibits its translocation to mitochondria, while GRX1 reverses this process. The S-glutathionylation of PARKIN can be reversed by GRX1, while the mutated PARKIN-mediated mitophagy decreased the function of ATP synthase (Complex V) and this can be reversed by GSTo, which increases its glutathionylation in normal levels. In ALS, the S-glutathionylation of SOD1 triggers stable dimer dissociation and misfolding reactive monomers. This process can be reversed by GRX1. SOD1’s toxic aggregates interact with BCL2, and this leads to VDAC inhibition and the ablation of ATP production. SOD1 and CCS can be trapped in the mitochondria’s matrix by disulfide dimerization induced by MIA40, leading to the aggregation of SOD1 inside mitochondria, which, in turn, activates SOD2. GRX2 can reverse SOD1 oligomerization. In FRDA, frataxin deficiency leads to iron accumulation inside the mitochondria and inhibits heme synthesis and iron–sulfur cluster (ISC) biogenesis. The S-glutathionylation of ETC and TCA-cycle enzymes inhibits mitochondria respiration. Increased levels of ROS, iron accumulation, decreased levels of reduced glutathione and the downregulation of NRF2 and GPX4 trigger ferroptosis as the main cell death mechanism in FRDA. Created with BioRender.com.

**Figure 7 ijms-23-15849-f007:**
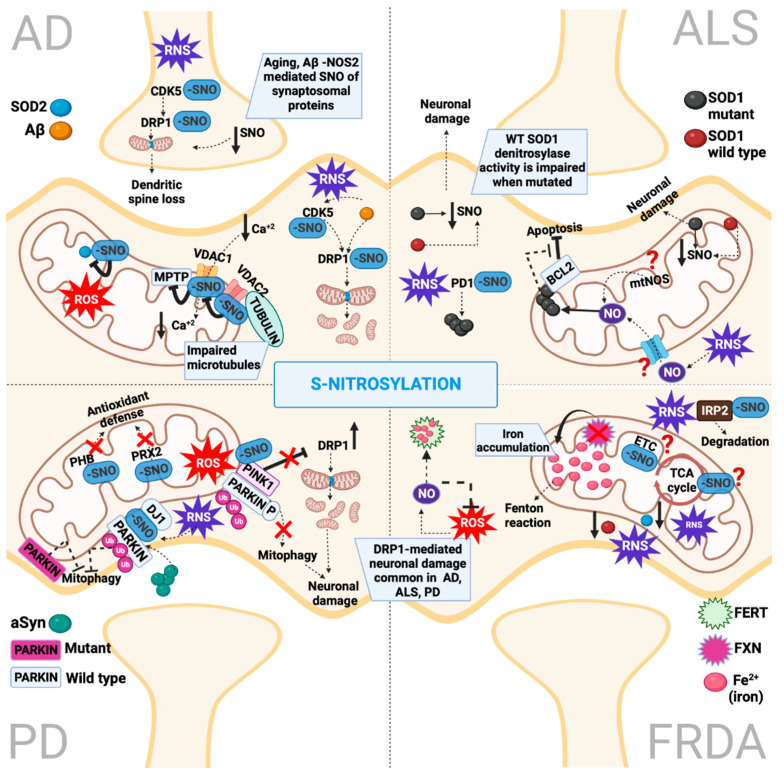
S-Nitrosylation impairs mitochondria homeostasis in AD, PD, ALS and FRDA. In AD, RNS mediated the S-nitrosylation of CDK5, which transnitrosylates DRP1, triggering excessive fission and leading to neuronal spine loss. The same mechanism can be initiated by Aβ oligomers. The S-nitrosylation of VDAC2 inhibits its function as well as VDAC1 function, hindering Ca^+2^ influx inside mitochondria. VDAC2 S-nitrosylation impairs microtubule architecture and further damages neurons. SOD2 S-nitrosylation impairs its detoxifying activity. The NOS2-mediated S-nitrosylation of various mitochondrial proteins in synaptosomes impairs synapses during aging and Aβ oligomerization. In PD, the S-nitrosylation of PARKIN suppresses its inhibitory function against DRP1 activation, leading to excessive mitophagy and neuronal damage. Moreover, it impairs its interaction with PINK1, preventing damaged mitochondrial clearance. The S-nitrosylation of PARKIN needs DJ1 stratification; this modification leads to the ubiquitination of PARKIN, preventing the mitophagy process. The same effect is caused by mutant PARKIN. Other mitochondrial proteins that are S-nitrosylated in PD are prohibitin and peroxiredoxin 2, which have inhibited antioxidant activities due to S-nitrosylation. In ALS, mutated SOD1 loses its denitrosylase activity and leads to decreased S-nitrosothiols levels in cytosol and mitochondria, which impairs the overall cellular homeostasis. The S-nitrosylation of PDI is implicated in mutant SOD1 oligomerization. SOD1 oligomers inhibit the anti-apoptotic function of BCL2, leading to apoptosis. Information is still scarce with respect to the origin of NO inside mitochondria and whether it triggers SOD1 oligomerization. NO either enters mitochondria via a denitrosylase transporter in mitochondrial membranes or is produced inside mitochondria by a local isoform of NOS. In FRDA, frataxin deficiency and iron accumulation lead to a Fenton reaction and excessive ROS. In turn, increased NO production is triggered as a ROS-scavenging mechanism, which also enhances iron accumulation inside ferritin. Aberrant NO-mediated S-nitrosylation decreases SOD1 and SOD2 activities, impairing their antioxidant activities. The S-nitrosylation of IRP2, which regulates iron accumulation, leads to its degradation. While ETC and TCA-cycle enzymes can be nitrosylated, it is still uncertain whether the increased NO production in FRDA also impairs mitochondrial respiration via this modification. Created with BioRender.com.

**Figure 8 ijms-23-15849-f008:**
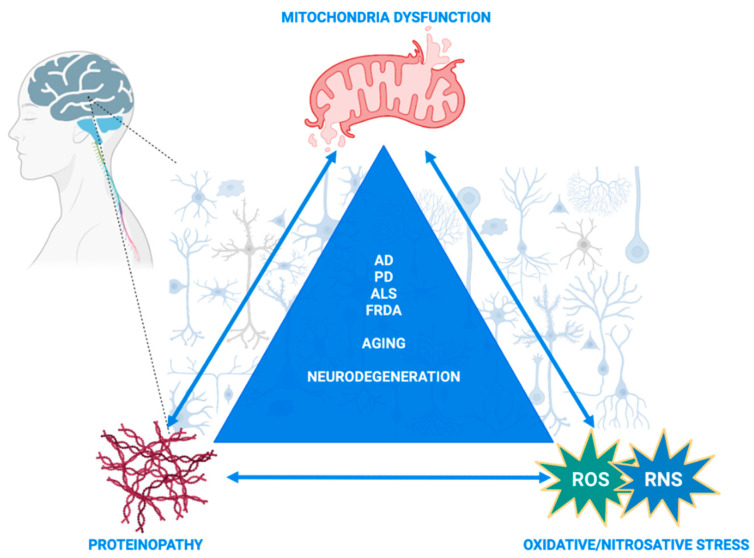
Trigonal interaction of ROS/RNS, mitochondria dysfunction and proteinopathy in aging and neurodegenerative disorders. In AD, PD, ALS and FRDA, mitochondria dysfunction is accompanied by increased oxidative/nitrosative stress, which in turn can trigger protein misfolding. Toxic protein aggregates such as Aβ, tau, a-synuclein and SOD1 can impair mitochondria homeostasis and further trigger ROS/RNS. Frataxin deficiency leads to mitochondria dysfunction and increased NO production, while the NO-mediated S-nitrosylation of iron-regulating proteins further increases iron accumulation and ROS/RNS production. Hallmark genes in AD, PD and ALS can be mutated and their effects on mitochondria homeostasis are phenocopied by redox modifications of the wild-type proteins. Created with BioRender.com.

**Table 1 ijms-23-15849-t001:** S-Glutathionylated (PSSG) & S-Nitrosylated (PSNO) mitochondrial proteins or proteins interacting with mitochondria in AD, PD, ALS and FRDA.

Protein	Biological Function	PSSG	PSNO	Disorder	PSSG/PSNO Effect	Ref.
α-Ketoglutarate dehydrogenase (KGDH) (E2 subunit)	Catalyzes the conversion of α-ketoglutarate to succinyl-CoA producing NADH directly providing electrons for the respiratory chain.	+		AD	Impairs glucose utilization. Decreases ATP & ROS production. Decreases rate of mitochondria respiration.	[132,142]
human Branched- chain aminotransferase protein (hBCAT) (CXXC motif)	Catalyzes reversible transamination of the α-amino group of the branched-chain amino acids to α-ketoglutarate, forming their respective branched chain α-keto acids and glutamate.	+		AD	Colocalizes with MIA40 & PDI in mitochondria. Role in Aβ misfolding.	[138,139]
Aβ (Not mitochondrial) (M35 residue)	Highly oxidized residue of Aβ which affects Aβ conformation.	+		AD	Lipid peroxidation, formation of amyloid plaques & neurofibrillary tangles. Also, correlated with mitochondria dysfunction.	[136]
Tau (Not mitochondrial) (C-terminal microtubule-binding region)	Microtubule-associated protein, forms insoluble filaments that accumulate as neurofibrillary tangles in AD.	+		AD	Increases tau dimerization & mitochondria dysfunction.	[140,141]
Mn superoxide dismutase (SOD2)	Manganese superoxide dismutase is the essential mitochondrial antioxidant enzyme that detoxifies the free radical superoxide, the major by-product of mitochondrial respiration.		+	AD	Inhibit its detoxifying capacity leading to mitochondrial dysfunction	[143]
Voltage-dependent anion-selective channel protein 1 (VDAC1)	VDACs promote mitochondrial transport of calcium ions. Part of the mitochondrial permeability transition pore (MPTP), facilitate cytochrome c release, leading to apoptosis. Interact with pro- & anti-apoptotic proteins at the outer mitochondrial membrane.		+	AD	Impaired Ca^+2^ transfer to mitochondria. Decreased ATP levels.	[30,144]
Voltage-dependent anion-selective channel protein 2 (VDAC2)			+	AD	Interaction TUBA-1A, 1B chain and TUBB-2C leading to impaired microtubes architecture. Impaired Ca^+2^ transfer to mitochondria.	
Dynamin-related protein 1 (DRP1) (Cys644)	Facilitates fission, promoting cytochrome c release and apoptosis.		+	AD	Increases mitochondria fragmentation leading to bioenergetics deficits & neuronal damage.	[145,146]
Cyclin-dependent kinase 5 (CDK5) (Cys83, Cys157)	Proline-directed serine/threonine-protein kinase essential for neuronal cell cycle arrest and differentiation. It is involved in apoptotic cell death in neuronal diseases (AD, PD) by triggering abortive cell cycle re-entry.		+	AD	Triggers Aß-mediated dendritic spine loss and neuronal damage. Transnitrosylates Drp1 increasing mitochondria fragmentation.	[147]
Succinate dehydrogenase (ubiquinone) flavoprotein subunit	Essential for assembly & activity of succinate dehydrogenase (TCA cycle).		+	AD	Unknown effect *	[148]
Succinyl-CoA ligase (ADP-forming) subunit beta	Essential for assembly & activity of succinate synthase (TCA cycle).		+	AD	Unknown effect *	[148]
Acyl carrier protein (ACP)	Co-factor of fatty acid biosynthesis.		+	AD	Unknown effect *	[148]
Succinyl-CoA: 3-ketoacid coenzyme A transferase 1	Transfers coenzyme A (CoA) from a donor thiol ester species (succinyl-CoA) to an acceptor carboxylate (acetoacetate), and produces acetoacetyl-CoA which is further metabolized to enter TCA cycle.		+	AD	Unknown effect *	[148]
NFU1 iron–sulfur cluster scaffold homolog	Critical of iron–sulfur cluster biogenesis.		+	AD	Unknown effect *	[148]
Pyruvate carboxylase (PC)	Catalyzes the conversion of pyruvate to oxaloacetate replenishing TCA cycle intermediates. Participates in gluconeogenesis, lipogenesis & neurotransmitter synthesis.		+	AD	Unknown effect *	[148]
Complex I (CI) (75-kDa subunit)	Constituent of electron transport chain. First rate-limiting enzyme.		+ **	PD	Inhibits mitochondrial respiration. Induces mitochondria damage and neuronal death.	[149]
PARKIN (RING & IBR domains)	Ubiquitin E3 ligase which is stratified by PINK1 in outer mitochondrial membrane to promote mitophagy.		+	PD	Mitochondrial dysfunction, protein misfolding and ubiquitin-proteasome system (UPS) impairment. SNO effect inhibits its activity to suppress DRP1-mediated fission. DJ1 activation is essential for PARKIN-SNO.	[150,151,152]
ATP synthase (β subunit)	Part of membrane-bound ATP synthase complex. Role in catalytic sites.	+		PD	Impaired mitochondrial function.	[153]
Protein deoxyglycase (DJ1) (Cys53, Cys106)	Multifunctional protein: chaperone, scavenger of ROS, regulator of transcription & cell signaling. Its gene *PARK7* is mutated in familial PD.	+		PD	Increased proximity with CI. Increase apoptosis by impairing BCL- xL function in outer mitochondrial membrane	[154,155,156]
PTEN-induced kinase 1 (PINK1) (Cys568)	Mitochondrial-targeted serine/threonine-protein kinase encoded by the *PINK1* gene which is mutated in familial PD. Protects from ROS by stratifying PARKIN & triggering mitophagy.		+	PD	Inhibits its kinase activity impairing PINK1/PARKIN-mediated mitophagy leading to dopaminergic neuronal cell death	[157]
Myocyte enhancer factor-2 (MEF2) (Not mitochondrial) (Cys39)	Transcriptional factor with key role in development of multiple organs.		+	PD	Inhibits MEF2-PGC1a transcriptional network, resulting in mitochondrial dysfunction and apoptosis.	[158]
Peroxiredoxin 2 (PRX2) (Cys51, Cys172)	Thiol-specific peroxidase that catalyzes the reduction of hydrogen peroxide & organic hydroperoxides to water and alcohols, accordingly.		+	PD	Diminishes peroxidase activity causing hydrogen peroxide to accumulate, exacerbating oxidative stress.	[159]
Prohibitin (PHB)	Mitochondrial chaperone protein		+	PD	Enhances its neuroprotective roles against ROS & glucose deprivation stress.	[160]
alpha-Synuclein (α-Syn) (Tyr39)	Pre-synaptic neuronal protein implicated in familial and sporadic PD pathogenesis.		**	PD	α-Syn nitration can potentiate a-synuclein oligomer formation. Extracellular α-Syn oligomers induce ROS/RNS-mediated nitrosylation of PARKIN leading to impaired mitophagy.	[161,162]
Cu-Zn Superoxide dismutase (SOD1) (mutated & wild type) (Cys111)	Major cytosolic antioxidant enzyme (with denitrosylase activity). It has been found in mitochondrial matrix. It is genetically & neuropathologically implicated in AD, PD and ALS.	+		ALS	Triggers SOD1 dimer disassembly, aggregation, loss of activity & neuronal damage. Triggers conformational changes in BCL2 & thus cytochrome c release and eventually apoptosis.	[163]
Protein disulfide isomerase (PD1) (Trx-like catalytic domain CXXC)	Multifunctional protein which associates with SOD1 misfolding. It is genetically & neuropathologically implicated in ALS.	+	+	ALS	Inhibits its activity, triggers mutant SOD1 aggregation and increases neuronal cell death	[142]
α-Ketoglutarate dehydrogenase (KGDH) E2 subunit	Catalyzes the conversion of α-ketoglutarate to succinyl-CoA producing NADH directly providing electrons for the respiratory chain. Constituents of electron transport chain.	+	? ***	FRDA	Its glutathionylation limits the production of NADH and the electron flow in the respiratory chain	[55,164]
Complex III, Complex IV	Maintains cellular iron homeostasis. It is required for mitochondrial iron supply & function.	+	? ***	FRDA	Impairs mitochondrial respiration.	[164]
Iron regulatory protein 2 (IRP2) (Cys178)			+	FRDA	Malfunction of iron homeostasis through UPS-dependent degradation of IRP2 that results in increased accumulation of iron inside the iron storage protein ferritin.	[165,166]

* These mitochondrial NOS2-dependent S-nitrosylation targets were identified by the proteomic analyses of synaptosomes derived from aged or APP/PS1 mice. The effect of S-nitrosylation on these proteins and their roles in AD pathogenesis is still elusive. ** Complex I can be both S-nitrosylated and nitrated, whereas α-synuclein can only be modified by nitration. *** It has not yet been elucidated whether these proteins are affected by NO-mediated S-nitrosylation in FRDA.4.1.4. GSTs Diverse Roles in Alzheimer’s Disease.

## Data Availability

Not applicable.

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
