# Peer review of "S-Glutathionylation and S-Nitrosylation in Mitochondria: Focus on Homeostasis and Neurodegenerative Diseases"

_ijms, 2022, doi:10.3390/ijms232415849_

Round 1

Reviewer 1 Report

I have some comment on the manuscript entitled “S-Glutathionylation and S-Nitrosylation in Mitochondria: Focus on Homeostasis and Neurodegenerative Diseases”.

1.       Write limitations of your manuscript at the end of the abstract section.

2.       More signalling components should be included in figure no 1.

3.       Elaborate your major objectives and approaches in the last paragraph of the introduction section.

4.       Improve the clarity of figure no 2.

5.        Limitations and future prospective should be included in the conclusion section.

6.       Discuss the Anti-Parkinsonian activity of Mucuna pruriens, ursolic acid and chlorogenic acid with respect to glutathione in MPTP intoxicated mouse model in Parkinson’s disease section.

7.       Mechanism of action with respect to upstream and downstream regulator should be included in the table.

8.       Complete editorial checking will be needed for the manuscript.

Author Response

  • ‘More signaling components should be included in Figure 1’

Answer: The rationale behind the design of the 8 figures of this review was to start as simple as possible by initially showing the potential effects of redox PTMs, then to describe the enzymatic processes of S-glutathionylation/S-nitrosylation (figure 2 and 4) and after to start giving more details about signaling pathways from mitochondria homeostatic mechanisms (figure 3 and 5) to more complex and multiple signaling pathways between disorders (figure 6 and 7). And at the end, to conclude with another simple illustration as in the beginning (figure 8). While we understand that more signaling pathways could be illustrated in figure 1, it would need details for all reversible and irreversible oxidative modifications and that would deviate from what we want to potentiate in this review (only S-glutathionylation and S-nitrosylation). We believe keeping simple the first and the last figure would help the reader gradually enter the extensive information and signaling pathways we depict in the rest of the figures.

  • ‘improve clarity of figure 2’

Answer: The reviewer does not specify whether the clarity concerns the illustration or the scientific content. Given that figure 2 and figure 4 have been made the same way, meaning that they address the components of S-glutathionylation/S-nitrosylation and the effect of these aberrant modifications in physiology and disease, we will keep them as they are. We decided to depict these complicated PTM’s mechanisms and their effect as simply as possible in order the reader, whether comes from a redox background or not, to quickly grasp the mode of action and the major components that constitute these mechanisms.

  • ‘Mechanism of action with respect to upstream and downstream regulator should be included in the table’.

Answer: In the table we gathered in 7 columns and 3 pages information about the targets of S-glutathionylation and/or S-nitrosylation stating whether they are mitochondrial or not and explaining both the target’s physiological roles but also the biological effect of the redox PTMs. We believe this is adequate information for the reader to understand what the target is, which targets are also associated with mitochondria deregulation through redox PTMs as well as the effect of these modifications. Concerning what is the upstream or downstream regulator, in the cases of the disorders discussed, it is still vague what is the trigger or exacerbator and it falls under the different interpretation of the researchers produced these results occasionally using different cell model systems of the disorders. To make the table more precise we added the exact residues and or subunits that are modified by these redox PTMs where possible.

  • ‘Elaborate your major objectives and approaches in the last paragraph of the introduction section’.

Answer: While we have already addressed the goal of this review and pinpoint the choice of the disorders, given that the reviewer addresses that additional information is needed for the reader to understand better the major objectives and approaches, we elaborated more about these points at the last paragraph of introduction lines 112 to 120.

  • ‘Limitations and future prospective should be included in the conclusion section’.

Answer: In the conclusions section, we have clearly addressed that in future redox proteomics will shed light on novel proteins that are modified by redox PTMs and that the new era of treatment will encompass molecules that target redox PTMs directly. However, we had not included limitations of this review which we did in accordance with the reviewer’s suggestion but at the end of the introduction section.

  • Write limitations of your manuscript at the end of the abstract section’.

Answer: While we understand that limitations of this review should be addressed, we believe it is more appropriate to mention them once at the end of the introduction section to make the reader aware of what to expect before reading this review.

  • Complete editorial checking will be needed for the manuscript.

Answer: Indeed, the manuscript will be edited appropriately by a professional service.

  • ‘Discuss the Anti-Parkinsonian activity of Mucuna pruriens, ursolic acid and chlorogenic acid with respect to glutathione in MPTP intoxicated mouse model in Parkinson’s disease section’.

Answer: Flavonoids are promising compounds in Parkinsonian models and in other models of disorders which include oxidative stress in their manifestation. However, in this review we wanted to exemplify proteins that are modified by S-glutathionylation or S-nitrosylation and represent either hallmarks of the disorders discussed or pivotal proteins for mitochondrial homeostasis. For Mucuna pruriens, ursolic acid and chlorogenic acid, important research has been done addressing their anti-Parkinsonian activity, but the mechanism of action is the Glycogen synthase kinase-3 (GSK-3) reduction of expression coupled with calcium inhibition, apoptosis and mitochondrial dysfunction W. Zahra et al., Phytomedicine Plus 2 (2022), Rai, SN. et al., Neurotox Res 36, 452–462 (2019), Rai SN. et al., Aging Neurosci. (2017). While these compounds show promising antioxidant and anti-inflammatory response, they do not act through modifying redox PTMs in proteins involved at least in mitochondria, or it hasn’t been shown yet to our knowledge. In the particular case of chlorogenic acid in which it has been shown to positively affect mitochondria complex I, IV and V activities, its mode of action is rather indirect by alleviating the increased ROS formation which might hinder mitochondrial complexes and through modifying Akt, ERK1/2, and GSK3β signalling pathways Singh et al., OxidMed and CelLong (2020). This is the reason why we will not mention them. The positive impact their administration has in glutathione levels cannot imply that these compounds are modifying proteins via S-glutathionylation, so their mode of action is not relevant to the purpose of this review.

Author Response

  1. All the figures are nicely depicted but the legends for all these figures are too lengthy and need to be crisp.

Answer: We decided to include adequate information at the legends to make it more convenient to the reader to understand each figure without the need to go through the text multiple times. Especially in the case of figures 3, 5, 6 & 7 we gathered information from multiple sections of the manuscript and decided to include the description of the signaling pathways to assist in the better understanding of the figures with the least effort.

  1. . Concerning the ETC and TCA cycle, these enzymes include aconitase, a-ketoglutarate dehydrogenase. a or α??

Answer: Concerning the alpha-ketoglutarate dehydrogenase (α-KGDH), the correct abbreviation is with alpha (α), while the whole enzyme can be stated either as alpha-ketoglutarate dehydrogenase complex (KGDH) or as oxoglutarate dehydrogenase complex (OGDC) https://www.uniprot.org/uniprotkb/P82909/entry. Given that the reviewer pinpointed that this abbreviation was not correctly addressed, we carefully corrected throughout the manuscript this mistake.

  1. Alzheimer’s disease (AD) is a progressive neurodegenerative disorder and the most prevailing form of dementia. AD is characterized by progressive loss of memory and  cognitive function and brain changes that include brain atrophy, amyloid-beta (Aβ) peptide build up, extracellularly, hyperphosphorylated tau protein, which forms neurofibrillary tangles (NFTs), intracellularly, neuronal death and inflammation. Adding latest literature will aid in the improvement https://doi.org/10.3390/biom9090495

Answer: Indeed, adding latest literature is the most appropriate choice and for this reason we added Matej et al., Clinical Biochemistry (2019) at line 457. The paper that the reviewer proposed as an alternative to the previous citation we had included, it is a research article which focuses on and briefly describes iron homeostasis deregulation in multiple neurodegenerative disorders, including AD. While there are indications for iron in AD, this does not belong to the major hallmarks of the disorder, so we will not include this but a more focused on clinical hallmarks of AD review (mentioned above).

  1. During aging, oxidative stress is increased in the brain due to the imbalance of the redox status, which includes the production of excess ROS or the deregulation of the antioxidant defence system. Accruing evidence has readily proposed that oxidative stress can temporally precede the onset of AD pathogenesis . Cite a relevant reference.

Answer: For the first sentence we cited at line 483 Liguori et al., Clinical Interventions in Aging (2018). For the second sentence we cited at line 484 Greenough et al., Neurochem Int. (2013) according to the reviewer’s suggestion.

  1. eases energy production and a-Crystallin B. Check for proper usage of symbols.

Answer: Indeed, the proper abbreviation is α-Crystallin B, and we corrected it where it was improperly written.

  1. Role of Glutathione in Aβ and Tau Accumulation in Alzheimer’s Disease. This section needs to be updated as it is written in a vague manner.

Answer: Based on the reviewer’s concern, we modified this section (lines 579-609), to improve the clarity of the information given.

  1. Parkinson’s disease (PD) is a neurodegenerative disorder clinically characterized by resting tremor, rigidity, postural instability and bradykinesia. Reference?

Answer: We added Bloem BR. et al., Lancet. (2021) at lines 775 and 777.

  1. The English grammar need to be checked, at some places there are long sentences that need to be restructured.

Answer: The manuscript has been edited for its English grammar by a professional service offered by the MDPI publisher. Concerning the long sentences, we will also restructure them, where possible, without changing the overall meaning.

  1. The manuscript should be proofread for erratic usage of abbreviations at first hand.

Answer: The manuscript was carefully revised to address potential erratic usage of abbreviations according to the reviewer’s suggestion.

Reviewer 3 Report

This is an extensive review on the role of S-glutathionylation and S-Nitrosylation in mitochondria with a focus on homeostasis and neurodegenerative diseases. The review has a potential for high impact, however, I have the following major suggestions that the authors may consider.

Concerns:

1.       Some statements are made but without references to back them, can the authors please add reference to the statement on PTM extensively studied but no references backing this statement lines 77&78 and statements in lines 79 & 80

2.       Can the authors state the mechanisms of non-enzymatic S-glutathionylation

3.       Certain words are first mention without defining their meaning such as nutoposis and mitoposis

4.       Can the authors elaborate more on what the meant by neuronal necroptosis?

5.       How does S-glutathionylation facilitate metabolic adaptation through fusion?

6.       Give examples of NO and NO derivatives that plays role in signaling

Author Response

  • ‘Some statements are made but without references to back them, can the authors please add reference to the statement on PTM extensively studied but no references backing this statement lines 77&78 and statements in lines 79 & 80’.

Answer: In the introduction at line 83 we added the appropriate literature that was missing since this will be indeed more appropriate. The following lines give a brief description of the topics we will cover for which bibliography has been extensively cited throughout the review, so we will not add additional references in those lines at the end of introduction.

  • ‘Can the authors state the mechanisms of non-enzymatic S-glutathionylation’.

Answer: In this review we aimed to focus on S-glutathionylation and S-nitrosylation modifications of proteins that affect mitochondrial homeostasis as well as implications in these two mechanisms in neurodegenerative diseases. Since aberrant S-glutathionylation has been associated with problematic enzymes that regulate this process and are also the focus of investigation in the disorders mentioned, we prefered to describe and illustrate the enzymatic process to minimize the biochemical details for the reader. Given that S-glutathionylation, enzymatic and non enzymatic, has been extensively reviewed, we simply cited the appropriate references for the reader who wishes to dive into more details. However, the absence of description of the multiple ways of non enzymatic S-glutathionylation will not prevent the reader to comprehend the topics covered in this review.

  • ‘Certain words are first mention without defining their meaning such as nutoposis and mitoposis’

Answer: In this review we have mentioned necroptosis and mitoptosis. We believe the reviewer was mentioning to these two words. We understand that according to the reviewer it would be more appropriate to briefly describe those when first mention them and we will modify the text accordingly. Concerning mitoptosis, which is primed by MPTP opening and ROS, we added additional explanation in lines 261 to 268. As it concerns neuronal necroptosis, we elaborated more about the exact mechanism at lines 319-326.

  • ‘Can the authors elaborate more on what the meant by neuronal necroptosis?’

Answer: According to the reviewer’s suggestion, we elaborated more about this mechanism at lines 319-326.

  • How does S-glutathionylation facilitate metabolic adaptation through fusion?

Answer: In lines 316-319 we added additional explanation for the hyperfusion state S-glutathionylation promotes and its effect in energy production.

  • Give examples of NO and NO derivatives that plays role in signalling

Answer: In lines 289&290 where is mentioned the NO and NO derivatives roles in modification of proteins, we added according to the reviewer’s suggestion examples of reactive nitrogen species generated by reaction of NO with other free radicals that in turn, modify proteins through covalent post-translational modifications (lines 299 to 304).

Reviewer 4 Report

This is potentially interesting literature review. Its authors, lacking own research results in the field of mitochondrial S-glutathionylation and S-nitrosylation (as evidenced by the absence of their papers in the list of references), consider existing literature data on this subject in an attempt to demonstrate the role of these processes in homeostasis and neurodegeneration. The literature contains 280 references, including only 10 references of 2021-2022 (!) Particular targets of mitochondrial S-glutathionylation and S-nitrosylation have not been characterized in details. For example, considering a-ketoglutarate dehydrogenase (KGDH) complex, consisting of several enzymes, as a target for S-glutathionylation/S-nitrosylation, the author do not indicate, which component(s) underwent these modifications. In this context, results of a recent report indicate one particular mechanism (Seim, G.L., John, S.V., Arp, N.L. et al. Nitric oxide-driven modifications of lipoic arm inhibit α-ketoacid dehydrogenases. Nat Chem Biol (2022). https://doi.org/10.1038/s41589-022-01153-w).

The same criticism may be addressed to proteins listed in Table 1. In some cases, the authors indicate a crucial residue (e.g. Abeta M35), or a certain protein subunit (e.g. Succinate dehydrogenase (ubiquinone) flavoprotein subunit or ATP synthase β subunit) while in other cases they just indicate Complex I (CI), which consists of many protein subunits. I believe that these details (where possible) will certainly improve this interesting review.

Author Response

  • “This is potentially interesting literature review. Its authors, lacking own research results in the field of mitochondrial S-glutathionylation and S-nitrosylation (as evidenced by the absence of their papers in the list of references), consider existing literature data on this subject in an attempt to demonstrate the role of these processes in homeostasis and neurodegeneration. The literature contains 280 references, including only 10 references of 2021-2022 (!) Particular targets of mitochondrial S-glutathionylation and S-nitrosylation have not been characterized in details. For example, considering a-ketoglutarate dehydrogenase (KGDH) complex, consisting of several enzymes, as a target for S-glutathionylation/S-nitrosylation, the author do not indicate, which component(s) underwent these modifications. In this context, results of a recent report indicate one particular mechanism (Seim, G.L., John, S.V., Arp, N.L. et al.Nitric oxide-driven modifications of lipoic arm inhibit α-ketoacid dehydrogenases. Nat Chem Biol (2022). https://doi.org/10.1038/s41589-022-01153-w). The same criticism may be addressed to proteins listed in Table 1. In some cases, the authors indicate a crucial residue (e.g. Abeta M35), or a certain protein subunit (e.g. Succinate dehydrogenase (ubiquinone) flavoprotein subunit or ATP synthase β subunit) while in other cases they just indicate Complex I (CI), which consists of many protein subunits. I believe that these details (where possible) will certainly improve this interesting review.”

Answer:

Here we reviewed the literature concerning the targets of S-glutathionylation and S-nitrosylation in mitochondria homeostatic mechanisms and in neurodegenerative disorders, where S-glutathionylation/S-nitrosylation has been implicated directly or indirectly in mitochondria deregulation. We also aimed to discuss the events in Friedrich’s ataxia which despite it’s a mitochondria disorder with evidence of glutathione and nitric oxide levels deregulation, it has not been included in relevant reviews. The field of redox biochemistry is small and the last twenty years only a small list of proteins has been addressed to be S-glutathionylated/S-nitrosylated, let alone that only a subfraction of those targets has been implicated in disease manifestation or exacerbation. We sought to bring together the knowledge existed so far but to compare these two mechanisms not only in mitochondria homeostasis but also between neurodegenerative disorders to inspire researchers who work in these specific disorders to look deeper on how the redoxome could interfere with disorders resembling aging. By scrutinizing the literature, we found 38 papers of the last 5-6 years in which already known targets of redox PTMs have been studied either in the context of homeostasis of mitochondria or for their implication in disease. We tried to include the most recent discoveries concerning this matter at almost every section of this review.

While we have not published redox-related work yet, we have initiated investigation on these redox PTMs in specific animal models of disorders that haven’t been investigated so far. We aimed to initiate this new chapter of investigation in our lab with a review that potentiates what we know so far and what we do not know.

Concerning the important suggestions of the reviewer for a more precise representation of redox PTMs targets, we added in the table and throughout the manuscript the residues or subunits that are S-glutathionylated/S-nitrosylated where possible. In addition, the reference the reviewer cited (Seim, G.L., John, S.V., Arp, N.L. et al. Nitric oxide-driven modifications of lipoic arm inhibit α-ketoacid dehydrogenases. Nat Chem Biol (2022). https://doi.org/10.1038/s41589-022-01153-w), is now included in the manuscript and commented at lines 423-438.

Round 2

Reviewer 1 Report

Revised version of the manuscript is not suitable for publication. 

Reviewer 3 Report

The authors have addressed all my concerns and I have no further concern on the manuscript.

Reviewer 4 Report

In my opinion, the revised version of the manuscript  now requires only editorial polishing and correction of some places (e.g. ref. 99 lacks information about a particular issue in which the cited paper has been published).